# Potential vorticity structure of embedded convection in a warm conveyor belt and its relevance for the large-scale dynamics

Annika Oertel[1], Maxi Boettcher[1], Hanna Joos[1], Michael Sprenger[1], and Heini Wernli[1]

[1]IAC, ETH Zürich, Zürich, Switzerland

**Correspondence:** Annika Oertel (annika.oertel@env.ethz.ch)

**Abstract.** Warm conveyor belts (WCBs) are important airstreams in extratropical cyclones. They can influence the large-scale flow evolution by modifying the potential vorticity (PV) distribution during their cross-isentropic ascent. Although WCBs are typically described as slantwise ascending and stratiform cloud producing airstreams, recent studies identified convective activity embedded within the large-scale WCB cloud band. Yet, the impacts of this WCB-embedded convection have not been investigated in detail. In this study, we systematically analyse the influence of embedded convection in an eastern North Atlantic WCB on the cloud and precipitation structure, on the PV distribution, and on the larger-scale flow. For this, we apply online trajectories in a high-resolution convection-permitting simulation and perform a composite analysis to compare quasi-vertically ascending convective WCB trajectories with typical slantwise ascending WCB trajectories. We find that the convective WCB ascent leads to substantially stronger surface precipitation and the formation of graupel in the mid- to upper troposphere, which is absent for the slantwise WCB category, indicating the key role of WCB-embedded convection for precipitation extremes. Compared to the slantwise WCB trajectories, the initial equivalent potential temperature of the convective WCB trajectories is higher and the convective WCB trajectories originate from a region of larger potential instability, which gives rise to more intense cloud diabatic heating and stronger cross-isentropic ascent. Moreover, the signature of embedded convection is distinctly imprinted in the PV structure. The diabatically generated low-level positive PV anomalies, associated with a cyclonic circulation anomaly, are substantially stronger for the convective WCB trajectories. The slantwise WCB trajectories lead to the formation of a wide-spread region of low-PV air (but still with weakly positive PV values) in the upper troposphere, in agreement with previous studies. In contrast, the convective WCB trajectories form mesoscale horizontal PV dipoles at upper levels, with one pole reaching negative PV values. On the larger-scale, these individual mesoscale PV anomalies can aggregate to elongated PV dipole bands extending from the convective updraft region, which are associated with coherent larger-scale circulation anomalies. An illustrative example of such a convectively generated PV dipole band shows that within around 10 hours the negative PV pole is advected closer to the upper-level waveguide, where it strengthens the isentropic PV gradient and contributes to the formation of a jet streak. This suggests that the mesoscale PV anomalies produced by embedded convection upstream organise and persist for several hours, and therefore can influence the synoptic-scale circulation. They thus can be dynamically relevant, influence the jet stream and potentially the downstream flow evolution, which are highly relevant aspects for medium-range weather forecast. Finally, our results imply that a distinction between slantwise and convective WCB trajectories is meaningful because the convective WCB trajectories are characterized by distinct properties, such as the formation of graupel and of an upper-level PV dipole, which are not present for the slantwise WCB trajectories.

## 1 Introduction

### 1.1 Warm conveyor belts and embedded convection

Moist diabatic processes are known to play an important role for the evolution of extratropical cyclones and are frequently associated with rapid cyclogenesis (e.g., Anthes et al., 1983; Kuo et al., 1991; Stoelinga, 1996; Wernli et al., 2002) and increased forecast error growth (e.g., Davies and Didone, 2013; Selz and Craig, 2015). Diabatic processes are particularly important in warm conveyor belts (WCBs), which are coherent, typically poleward ascending airstreams associated with extratropical cyclones (Harrold, 1973; Browning, 1986, 1999; Wernli and Davies, 1997). During their typically slantwise cross-isentropic ascent from the boundary layer ahead of the cold front to the upper troposphere, they form large-scale mostly stratiform cloud bands and play a key role for the distribution of surface precipitation (e.g., Browning, 1986; Eckhardt et al., 2004; Madonna et al., 2014; Pfahl et al., 2014; Flaounas et al., 2018).

Although WCBs are typically described as gradually ascending and mainly stratiform cloud producing airstreams (e.g., Browning, 1986; Madonna et al., 2014), already in 1993 the concept of rapid convective motion embedded in the frontal cloud band of the WCB was proposed (Neiman et al., 1993). Recent studies suggested that the WCB is, at least in some cases, not a homogeneously ascending airstream: in contrast, the detailed ascent behavior of the individual WCB trajectories associated with one extratropical cyclone can vary substantially (e.g., Martínez-Alvarado et al., 2014; Rasp et al., 2016; Oertel et al., 2019) and convective activity can be frequently embedded in the large-scale baroclinic region of the WCB. This has been identified, e.g., with various remote-sensing data (Binder, 2016; Crespo and Posselt, 2016; Flaounas et al., 2016, 2018), with online trajectories in convection-permitting simulations (Rasp et al., 2016; Oertel et al., 2019) and in coarser simulations with parameterized convection (Agustì-Panareda et al., 2005; Martínez-Alvarado and Plant, 2014).

The strong cloud diabatic processes in both WCBs and convection modify the potential vorticity (PV) distribution in the lower and upper troposphere and thereby can affect the larger-scale dynamics (e.g., Pomroy and Thorpe, 2000; Grams et al., 2011; Clarke et al., 2019). Hence, WCBs and convection can be associated with increased forecast uncertainty (Baumgart et al., 2019; Berman and Torn, 2019) and forecast errors (e.g., Martínez-Alvarado et al., 2016a; Clarke et al., 2019). Occassionally, convection and WCBs individually can be related to forecast busts (Rodwell et al., 2013; Grams et al., 2018). The specific PV signatures of (i) large-scale WCB ascent and (ii) smaller-scale convective updrafts, and their potential implications for the flow evolution differ substantially and are discussed in the following.

### 1.2 PV modification by WCBs and convection

PV is materially conserved along the flow only in the absence of friction and diabatic processes (Hoskins et al., 1985). Neglecting frictional processes, the Lagrangian rate of change $\frac{D}{Dt}PV$ can be expressed as

$$\frac{D}{Dt}PV = \frac{1}{\rho}\boldsymbol{\omega} \cdot \nabla \dot{\theta}, \tag{1}$$

where PV is defined as (Ertel, 1942)

$$PV = \frac{1}{\rho}\boldsymbol{\omega} \cdot \nabla \theta \qquad (2)$$

and $\rho$ is density, $\theta$ is potential temperature, $\dot{\theta}$ represents diabatic heating or cooling, and $\boldsymbol{\omega}$ is 3D absolute vorticity ($\boldsymbol{\omega} = \nabla \times \mathbf{u} + 2\boldsymbol{\Omega} = \xi\mathbf{i} + \eta\mathbf{j} + (f+\zeta)\mathbf{k}$, where $\mathbf{u}$ is the 3D wind vector, $\boldsymbol{\Omega}$ is the vector of earth rotation, $\xi$ and $\eta$ are the horizontal vorticity components in the x- and y-directions, $f$ is the Coriolis parameter, and $f + \zeta$ is the absolute vertical vorticity).

For large-scale and predominantly slantwise WCB ascent it is frequently assumed that the first-order effect of latent heating on PV is dominated by the vertical gradient of diabatic heating (e.g., Wernli and Davies, 1997; Joos and Wernli, 2012; Madonna et al., 2014) resulting in PV generation below and PV destruction above the diabatic heating maximum according to Eq. 3:

$$\frac{D}{Dt}PV \approx \frac{1}{\rho}(f+\zeta) \cdot \frac{\partial \dot{\theta}}{\partial z}. \qquad (3)$$

These diabatically produced low-level positive and upper-level negative PV anomalies can lead to cyclone intensification (Rossa et al., 2000; Binder et al., 2016) and modify the upper-level flow evolution (e.g., Pomroy and Thorpe, 2000; Grams et al., 2011; Schäfler and Harnisch, 2015; Joos and Forbes, 2016; Martínez-Alvarado et al., 2016b).

PV is frequently considered for synoptic-scale dynamics (e.g., Hoskins et al., 1985; Stoelinga, 1996), but is also suited for the analysis of mesoscale convective systems (e.g., Conzemius and Montgomery, 2009; Shutts, 2017; Clarke et al., 2019) and has already been applied at the scale of individual convective cells (Chagnon and Gray, 2009; Weijenborg et al., 2015, 2017). While for PV modification in synoptic-scale systems the horizontal gradient of diabatic heating is frequently neglected (as in Eq. 3), this assumption breaks down in the case of intense local diabatic heating such as embedded mesoscale convective updrafts on the scale of a few kilometers. On the mesoscale, the horizontal gradients of $\dot{\theta}$ become relevant and Eq. 3 generalizes to the full form (Eq. 2), written here as

$$\frac{D}{Dt}PV = \frac{1}{\rho}\left[(f+\zeta)\frac{\partial \dot{\theta}}{\partial z} + \boldsymbol{\omega_h} \cdot \nabla_h \dot{\theta}\right] \qquad (4)$$

where $\boldsymbol{\omega_h}$ denotes the horizontal vorticity ($\boldsymbol{\omega_h} = \xi\mathbf{i} + \eta\mathbf{j}$). Previous studies showed that the horizontal diabatic heating gradients play a major role for the PV modification in isolated convective updrafts (e.g., Chagnon and Gray, 2009; Weijenborg et al., 2015, 2017) and in narrow smaller-scale heating regions embedded in a larger-scale flow (Harvey et al., 2020). The localized diabatic heating in a vertically sheared environment generates upper-level horizontal PV dipoles centered around the convective updraft (Fig. 1) and aligned with the horizontal vorticity vector $\boldsymbol{\omega_h}$ (Eq. 4, Fig. 1, green arrow), which is rotated 90° anticlockwise to the vertical wind shear vector $\boldsymbol{v_z}$ (Fig. 1, black arrow). Thereby, the positive PV pole (red shading in Fig. 1) occurs to the right of the vertical wind shear vector $\boldsymbol{v_z}$ [since there, $\nabla_h \dot{\theta}$ (Fig. 1, grey arrows) is parallel to $\boldsymbol{\omega_h}$] and the negative pole (blue shading in Fig. 1) occurs to the left of $\boldsymbol{v_z}$ [where $\nabla_h \dot{\theta}$ and $\boldsymbol{\omega_h}$ are antiparallel]. The strongest PV production and destruction (Fig. 1, dark red and dark blue shading, respectively) arise where the horizontal vorticity vector (Fig. 1, green arrow) and the horizontal diabatic heating gradient, which points radially towards the center of the convective updraft

(Fig. 1, grey arrows), are quasi-aligned, i.e., where the angle $\alpha$ between both vectors is small, as $\frac{D}{Dt}PV \approx |\boldsymbol{\omega_h}| \cdot |\nabla_h \dot{\theta}| \cdot \cos \alpha$. The PV production and destruction is attenuated where the angle $\alpha$ between $\boldsymbol{\omega_h}$ and $\nabla_h \dot{\theta}$ increases (Fig. 1, orange and light blue shading, respectively). Thus, for smaller-scale diabatic heating, as in convective updrafts, the horizontal components of PV become increasingly dominant and generate an upper-level horizontal PV dipole, whereby one pole can reach negative PV values (Harvey et al., 2020).

This quasi-horizontal PV dipole structure is a robust response of convective updrafts in the presence of vertical wind shear (Chagnon and Gray, 2009; Weijenborg et al., 2015, 2017; Hitchman and Rowe, 2017). Such convectively generated PV dipoles were previously identified in idealized simulations of isolated cumulus-scale convection (Chagnon and Gray, 2009), in case studies of mesoscale convective systems (Davis and Weisman, 1994; Chagnon and Gray, 2009; Hitchman and Rowe, 2017; Clarke et al., 2019), and in mid-latitude convective updrafts with varying large-scale flow conditions (Weijenborg et al., 2015, 2017). The amplitude of the horizontal PV dipoles can strongly exceed the typical amplitude of synoptic-scale PV. In strong convective updrafts horizontal PV dipoles of $\pm 10$ PVU can be generated (Chagnon and Gray, 2009; Weijenborg et al., 2015, 2017), resulting in regions of absolute negative PV. These regions can be hydrostatically, inertially or symmetrically unstable (e.g., Hoskins, 2015) and can form mesoscale circulations associated, e.g., with frontal rainbands (Bennetts and Hoskins, 1979; Schultz and Schumacher, 1999; Siedersleben and Gohm, 2016), sting jets (Clark et al., 2005; Volonté et al., 2018, 2019), enhanced stratosphere-troposphere exchange (Rowe and Hitchman, 2015), and local jet accelerations and northward displacements (Rowe and Hitchman, 2016). The adjustment timescales for the release of these instabilities range from minutes for hydrostatic instability to several hours for inertial instability [timescale is proportional to $[-f(f + \zeta)]^{-1/2}$ (Schultz and Schumacher, 1999; Thompson et al., 2018)]. Thus, while hydrostatic instability is rapidly released and near-neutral conditions are established, inertial instability can prevail for several hours and therefore synoptic-scale regions of inertial instability can be found, for instance at the anticyclonic shear side of midlatitude ridges (Thompson et al., 2018).

Previous studies analysed the PV modification by individual convective updrafts and mesoscale convective systems. However, the PV modification by aggregated convection embedded in the WCB ascent region, which is already subject to strong diabatic PV modification from large-scale WCB ascent, has not yet been investigated. Hence, the contribution of embedded convection in WCBs to the distribution of PV and the formation of mesoscale PV anomalies, which may influence the development of extratropical cyclones, is still unknown. Moreover, the persistence and dynamical relevance of the convectively generated PV dipoles has not yet been analysed. Weijenborg et al. (2017) hypothesized that the convectively formed large-amplitude PV anomalies could be longer-lived than the relatively short-lived convective updrafts and suggested that a more detailed investigation of these PV anomalies might shed light on the dynamical relevance of convection. Related to this is the question whether the convectively generated PV dipoles aggregate to larger-scale PV anomalies and, if yes, whether they feed back on the synoptic-scale flow (Chagnon and Gray, 2009).

## 1.3 Aim and outline

In this study, we investigate convection embedded in a WCB case study and systematically analyse the PV modification of this convective activity. Furthermore, we go beyond the identification of convectively produced PV anomalies and evaluate the

effect of these mesoscale PV anomalies on the larger-scale flow, thereby emphasizing the dynamical relevance of embedded convection. Therefore, online trajectories in a high-resolution convection-permitting simulation are computed to compare convective and slantwise WCB trajectories and their impact on the cloud and precipitation structure, as well as on the mesoscale and larger-scale dynamics. Together, this study shows how WCB-embedded convection on the one hand influences the local mesoscale dynamics and on the other hand can modify the larger-scale flow evolution – both relevant aspects for predictability. Moreover, this study provides a refined perspective on the relevance of smaller-scale processes for the larger-scale WCB dynamics. Specifically, we address the following questions:

1. What is the impact of convective WCB ascent on the cloud and precipitation structure (section 3.2)?

2. What are characteristic thermodynamic properties of convective and slantwise WCB ascent (section 3.3)?

3. Where do convective and slantwise WCB trajectories originate from (section 3.4)?

4. How does convective (vs. slantwise) WCB ascent modify the PV distribution along the ascent and in its environment (section 3.5)?

5. What is the impact of convectively modified PV on the local wind speed and circulation in the upper troposphere (section 3.6)?

6. How does convection embedded in WCBs and its associated PV anomalies influence the larger-scale dynamics (section 4)?

This study is structured in the following way: Section 2 explains the methodology and shortly introduces the analysed WCB case study. Thereafter, we systematically consider the mesoscale effects of convection embedded in WCBs (questions 1-5) in a composite analysis (section 3). To address the question of the dynamical relevance of WCB-embedded convection (question 6), we consider an illustrative example of the characteristics and impact of WCB-embedded convection (section 4.1) and evaluate the influence of the convectively generated PV anomalies on the large-scale flow (section 4.2). Finally, we provide a discussion and outlook (section 5) and conclusions (section 6).

## 2 Data and approach

### 2.1 COSMO setup and trajectories

The WCB case study is simulated with the limited-area nonhydrostatic model COSMO (Consortium for Small-scale Modeling; Baldauf et al., 2011; Doms and Baldauf, 2018) at 0.02° (≈2.2 km) horizontal grid spacing with 60 vertical levels. The setup of the COSMO simulation and the online trajectories is the same as in Oertel et al. (2019). The simulation is initialized at 00 UTC 22 Sep 2016 in the early phase of the cyclogenesis of cyclone *Vladiana* (see section 2.2) and runs for 112 hours (see Oertel et al., 2019). Initial and lateral boundary conditions are taken from the ECMWF analyses with a horizontal resolution

of 0.1° every 6 hours. The domain is centered in the eastern North Atlantic and extends from about 50°W to 20°E and 30°N to 70°N. We apply the standard COSMO setup of the Swiss National Weather Service, which employs a one-moment six-category cloud microphysics scheme including prognostic water vapour ($q_v$), liquid (LWC) and ice (IWC) cloud water content, rain (LWC), snow (SWC) and graupel (GWC). The graupel category is important for the explicit simulation of deep convection (Baldauf et al., 2011). Deep convection is resolved at 2.2 km (e.g., Ban et al., 2014) while for shallow convection the reduced Tiedtke scheme was applied (Tiedtke, 1989; Baldauf et al., 2011). 3D COSMO fields are output every 15 minutes, which allows capturing the large temporal and spatial variability of embedded convection.

To identify phases of embedded convective ascent in the WCB, 10 000 online trajectories are started from a predefined region at 7 vertical levels (250, 500, 750, 1000, 1500, 2000 and 2500 m a.s.l.) every 2 h during the simulation. The online trajectory positions are calculated from the resolved 3D wind field at every model time step, i.e. every 20 s, and thus, explicitly capture rapid convective ascent (Miltenberger et al., 2013, 2014; Rasp et al., 2016; Oertel et al., 2019). WCB trajectories are subsequently selected as trajectories with an ascent rate of at least 600 hPa in 48 h (Madonna et al., 2014).

## 2.2 Overview of WCB case study

The investigated WCB is associated with the North Atlantic extratropical cyclone *Vladiana* that occurred between 22-25 Sep 2016 [IOP 3 of the North Atlantic Waveguide and Downstream Impact EXperiment (NAWDEX, Schäfler et al., 2018)]. The cyclone with a minimum sea level pressure of 975 hPa is located below an upper-level trough and propagates eastward and northward across the North Atlantic toward Iceland, where it becomes stationary (Fig. 2). The cyclone's WCB ascends in the warm sector predominantly in a narrow band ahead of the cold front and develops a weak cyclonic and a stronger anticyclonic branch (cf. Wernli, 1997; Martínez-Alvarado et al., 2014), which turns into the downstream upper-level ridge and contributes to its amplification (see Fig. 2d-f in Oertel et al., 2019). A more detailed analysis of the cyclone evolution and its WCB ascent is presented in Oertel et al. (2019).

The WCB trajectories in the baroclinic zone ahead of the cyclone's cold front vary considerably in their ascent rates (Fig. 2a-c; see also the animation in the online supplemental material) and also include phases of embedded convection. These phases are characterized by a rapid ascent of more than 400-600 hPa in 2 h, and they are embedded in a larger region of slower, more gradual WCB ascent (red circles in Fig. 2a-c and online supplemental material; cf. Oertel et al., 2019). The region of embedded convective activity is characterized by a very heterogeneous PV field of diabatically produced small-scale but high-amplitude PV anomalies of ±10 PVU in the upper troposphere (Fig. 2d-f and online supplemental material), suggesting that embedded convection in WCBs can strongly modify the PV distribution. There is, however, no clear separation between convective and slantwise ascent in the WCB of cyclone *Vladiana* (Oertel et al., 2019). Instead, it is rather a continuum of ascent rates ranging from very rapid convective ascent of more than 600 hPa in 2 h to a slower gradual ascent of approximately 50 hPa in 2 h. Nevertheless, we can meaningfully classify the WCB trajectories into two sub-categories based on their ascent rate to compare convective versus slantwise WCB ascent (see section 2.3).

## 2.3 WCB trajectory categorization and WCB ascent region

To compare the rapid convective WCB ascent to the "typical" slower and slantwise WCB ascent, we define two sub-categories of coherently ascending WCB trajectories: (i) convectively ascending WCB trajectories that perform a rapid quasi-vertical ascent through the whole tropospheric column, and (ii) slantwise WCB trajectories that ascend more slowly and gradually from the boundary layer into the upper troposphere. The selection criteria are based on the fastest 400-hPa and 600-hPa ascent phases along the WCB trajectories: a WCB trajectory is considered as convective if its fastest 400-hPa and 600-hPa ascent times are shorter than 1 h and 3 h, respectively. These ascent rates correspond to the 10% fastest ascent rates of all trajectories of the considered WCB. Likewise, a WCB trajectory is assigned to the slantwise WCB category if the 400-hPa and 600-hPa ascent times are between 1.5 h to 3.5 h and 6.5 h to 22 h, respectively. These ascent times correspond to the average ascent rates of all WCB trajectories ($25^{th}$ to $75^{th}$ percentiles). The selection criteria result in approximately 2000 convective WCB trajectories, and approximately 7000 more slowly ascending slantwise WCB trajectories.

Figure 3a shows the temporal evolution of the WCB trajectory positions at the start of their ascent relative to the approaching cold front at selected times. The main WCB ascent occurs ahead of the cold front and the upper-level trough (see also online supplemental material). The selected convective WCB trajectories perform a rapid and deep ascent through the whole troposphere mostly south of 50°N (Fig. 3a, black outlined circles), and the slantwise WCB trajectories with comparatively slow and gradual ascent rates are located ahead of, and travel northward with, the cold front during their ascent (Fig. 3a, grey outlined triangles). During the 3 days of major WCB ascent from 22 Sep to 24 Sep 2016, a joint evolution and eastward propagation of about 20° of the cyclone, its cold front and the WCB ascent region takes place (see colored symbols in Fig. 3a). Despite the differing ascent behavior of the WCB categories, the convective WCB ascent is directly embedded in the region of large-scale ascent, i.e., in close proximity to the more slowly ascending WCB trajectories. The convective WCB trajectories start their ascent on average slightly further south at the cold front (45.2° ±3°) compared to the slantwise WCB trajectories whose ascent region extends further poleward (47.7° ±4°). Nevertheless, the overall region of origin and ascent overlaps (Figs. 2a-c and 3a) and the convective WCB ascent is indeed embedded in the region of slower WCB ascent. This indicates that although their ascent rates differ, both WCB categories are not spatially separated (Fig. 3a).

## 3 Characteristics of convective and slantwise WCB ascent

### 3.1 Composite analysis

The similarities and differences of the characteristics of convective and slantwise WCB trajectories are systematically compared in a composite analysis. For computing composites for both WCB categories, the selected WCB trajectories are centered relative to the time of the start of the fastest 400-hPa ascent phase. Composites are computed based on the trajectory position every 15 minutes, which corresponds to the temporal resolution of the COSMO output.

Three types of composites are produced for both WCB categories: (i) composites of vertical profiles along the trajectories, i.e. time-height sections along the flow, and (ii) horizontal and (iii) vertical cross-sections centered at the trajectories' geo-

graphical position. While the along-flow composites provide a Lagrangian perspective on the local dynamical impact of the WCB trajectories, the horizontal and vertical cross-sections allow analysing the mutual interaction between the WCB and its surroundings. Because the trajectories are located in a region with coherent background flow ahead of the upper-level trough (cf. Fig. 2a-c), the fields are not rotated for the composite computation. This enables a direct interpretation of the atmospheric conditions and perturbations in geographic coordinates.

The number of selected convective and slantwise WCB trajectories is not homogeneous in time; instead pulses of convective and slantwise WCB ascent occur (Fig. 3a,b and cf. Oertel et al., 2019). In particular, two pulses of increased convective activity occur at around 00 UTC 23 Sep and 00 UTC 24 Sep. Hence, the composite analyses are dominated by these times when large numbers of WCB trajectories are selected for each category.

   The convective WCB trajectories ascend quasi-vertically from the boundary layer to the upper troposphere to on average
$10\,\mathrm{km}$ height ($\pm\,1.0\,\mathrm{km}$) in about 1-2 h (Fig. 4a, black line). In contrast, the slantwise WCB trajectories are characterized by a substantially slower ascent (in agreement with our selection criteria) and perform a gradual slantwise ascent until they reach their final outflow level at, on average, $9\,\mathrm{km}$ ($\pm\,1.2\,\mathrm{km}$) height after approximately 18 h, after an initially swift ascent (due to the centering relative to the fastest 400-hPa ascent) in the lower troposphere (Fig. 4b, black line). Thus, the final WCB outflow height of the slantwise WCB trajectories is on average lower than for the convective WCB trajectories.

In the following, we first describe the precipitation and cloud structure associated with the rapid convective and slower slantwise WCB trajectories (section 3.2). Subsequently, we analyse their thermodynamic properties (section 3.3) and compare the environment of both WCB categories (section 3.4). Then, the PV structure associated with convective and slantwise WCB ascent is investigated (section 3.5) and related to the flow anomalies (section 3.6).

## 3.2   Precipitation and cloud structure

The convective cloud formed during the rapid WCB ascent is characterized by large hydrometeor contents of up to $1\,\mathrm{g\,kg^{-1}}$ (Fig. 4a) and the vertically integrated rain, snow and graupel water path in close proximity to the updraft can reach up to $6\,\mathrm{kg\,m^{-2}}$ (Fig. 4e), forming a locally dense and vertically extended cloud (Fig. 4a). Directly above the convective updraft the cloud top height reaches a local maximum (Fig. 4a). During their rapid ascent the convective WCB trajectories locally produce intense surface precipitation (Fig. 4c,e). The precipitation maximum coincides with the strongest ascent phase in
the mid-troposphere (Fig. 4a,c), where also a local maximum of graupel production occurs (Fig. 4a, magenta contours). The maximum surface precipitation is slightly shifted upstream relative to the convective updraft (Fig. 4e). The upper-level WCB outflow remains inside a thick cirrus cloud for several hours, which has formed during the convective ascent and is subsequently advected with the upper-level mean flow (Fig. 4a, turquoise contours). This convectively formed cirrus cloud can be considered as a longer-lived convective anvil cloud, suggesting that embedded convection is also relevant for the larger-scale upper-level
cloud cover.

   The cloud formed during the slantwise WCB ascent is comparatively less dense, with lower rain, ice, and snow water content and without graupel production (Fig. 4b,f). Accordingly, the cloud structure and cloud top are more homogeneous and stratiform, and the surface precipitation maximum along the ascent (Fig. 4d) is substantially weaker compared to the

convective WCB ascent (peak value reduced by a factor of 3). The vertically integrated rain, snow and graupel water paths for
the slantwise WCB trajectories are substantially lower and distributed homogeneously over a larger area (Fig. 4f). The WCB
outflow is surrounded by a cirrus cloud during the entire ascent (Fig. 4b), indicating the relevance of WCBs for the formation
and maintenance of the extended upper-level cirrus cloud cover associated with extratropical cyclones (Eckhardt et al., 2004;
Madonna et al., 2014; Oertel et al., 2019; Joos, 2019). The denser cloud (Fig. 4a,b) with limited spatial extent (Fig. 4e,f) in the
convective case implies a pronounced heterogeneity of the large-scale cloud structure if convection is directly embedded in the
large-scale WCB cloud band.

A previous analysis showed that the precipitating region for the considered cyclone is spatially confined to the WCB ascent
region (cf. Oertel et al., 2019, Fig. 9b). Indeed, the (normalized) number of convective and slantwise WCB trajectories starting
their fastest ascent correlate both well with the evolution of the averaged (non-zero) precipitation in the WCB domain (Fig. 3b).
In particular, both convective ascent pulses clearly coincide with the domain-averaged precipitation maxima, suggesting that
the evolution of embedded convection in the WCB has an impact on the precipitation intensity (cf. Fig. 9 in Oertel et al., 2019).

The distinctly different cloud and precipitation structure between both WCB trajectory categories underlines the rationale
of our classification of convective versus slantwise ascent, and agrees with typical characteristics of both precipitation types:
The convective WCB ascent produces locally confined, intense precipitation including the formation of graupel, while the
precipitation associated with the slantwise WCB ascent is much less intense and distributed over a larger domain whereby the
ascent velocity is too slow for graupel production. Hence, the slantwise WCB ascent forms an extended stratiform cloud band.

### 3.3 Thermodynamic properties

Consistent with the formation of clouds and precipitation, the convective WCB trajectories experience substantial cloud dia-
batic heating of on average 35 K during the first 3 h, and they reach their outflow level at the 330 K isentrope in agreement
with their initial $\theta_e$ value of 330 K (Fig. 5a, cf. section 3.4). The averaged total cross-isentropic ascent of the slantwise WCB
trajectories with lower $\theta_e$ in the inflow is weaker with about 28 K in 18 h when their final outflow level is reached at around
323 K (Fig. 5b). The strong and localized heating in the convective updrafts leads to a local lifting of the melting level (Fig. 4a)
and a localized downward deflection of the isentropes in the diabatically heated region (Fig. 6c and 5a).

Both convective and slantwise WCB trajectories ascend only approximately along constant-$\theta_e$ surfaces (Fig. 5a,b) due to
the influence of ice microphysical processes during the ascent. The calculation of $\theta_e$ only considers the heat released during
the transition from the vapour to the liquid phase, but does not account for the additional heat release associated with the ice
phase [the transition from the vapour to the ice phase releases the latent heat of condensation plus freezing ($L_c + L_f$), and the
latter is not accounted for in the calculation of $\theta_e$]. Hence, the influence of melting from falling hydrometeors and the phase
transitions from liquid to ice above the $0°C$ isotherm are evident along the WCB ascent. Following the ascent of the convective
WCB trajectories, once the melting level is reached in the vicinity of the $0°C$ isotherm, i.e., where a transition from the solid
(SWC and GWC) to the liquid (RWC) phase occurs (Fig. 4c), $\theta_e$ decreases due to melting of snow and graupel falling into
the ascending air parcels (Fig. 5a). At higher altitudes, $\theta_e$ increases again due to the additional heat release in the ice phase
(Fig. 5a). This process of decrease and subsequent increase of $\theta_e$ along the ascent is also evident in the more slowly ascending

WCB trajectories (Fig. 5b), underlining the importance of microphysical processes in WCBs (Joos and Wernli, 2012; Joos and Forbes, 2016), whose effect is clearly detectable even after averaging over hundreds of trajectories. [1]

## 3.4 Environment for convective and slantwise WCB ascent

The rapidly ascending convective WCB trajectories originate from a slightly warmer and substantially moister region ($\theta = 296 \pm 1.3 \, \text{K}, q_v = 11.5 \pm 1.2 \, \text{g} \, \text{kg}^{-1}$) compared to the more slowly ascending slantwise WCB trajectories ($\theta = 295 \pm 1.9 \, \text{K}, q_v = 9.9 \pm 1.3 \, \text{g} \, \text{kg}^{-1}$), and are thus characterized by substantially higher initial $\theta_e$ (Figs. 6a and 5a) than the slantwise WCB trajectories (Figs. 6b and 5b). At the start of the ascent, $\theta_e$ amounts to 330 K ($\pm 4.8$ K) for the convective WCB trajectories and to 324 K ($\pm 5.9$ K) for the slantwise WCB trajectories (Fig. 5a,b). Although the convective ascent is embedded within the region of slantwise ascent ahead of the cold front (Figs. 3 and 6a), where $\theta_e$ contours nearly become vertical (Fig. 6c,d), the convective WCB trajectories ascend from a mesoscale, meridionally elongated region characterized by warmer and more humid conditions ahead of a strong localized $\theta_e$ gradient (Fig. 6a). This narrow tongue of very high $\theta_e$ air with $q_v$ exceeding 11 g kg$^{-1}$ extends laterally from ahead of the cold front only approximately 50 km into the warm sector and forms a strong horizontal $\theta_e$ gradient (Fig. 6a). Moreover, the WCB ascent region ahead of the cold frontal zone coincides with enhanced low-level convergence of the horizontal wind (Fig. 6c,d), which is particularly strong for the convective WCB trajectories. The mesoscale frontal $\theta_e$ structures ahead of the cold front arise from large $\theta_e$ variability in the warm sector. The higher $\theta_e$ of the convective WCB trajectories subsequently leads to more intense cloud diabatic processes and a faster and stronger cross-isentropic ascent (Fig. 5a,b).

The rapidly ascending, convective WCB trajectories (Fig. 5a) originate in the boundary layer from a region of strong potential instability characterized by vertical $\theta_e$ gradients of -4 K km$^{-1}$ prior to the start of the convective ascent (Fig. 5c). After their rapid ascent from the boundary layer into the upper troposphere, the convective WCB trajectories continue to moderately ascend almost isentropically along the upper-level ridge (Fig. 5a). The convective WCB ascent is likely triggered through lifting of the potentially unstable layer in the frontal ageostrophic circulation (cf. quasi-geostrophic omega ahead of cold front in Fig. 6 and section 5.1.2 in Oertel et al., 2019). During an initial adiabatic ascent the low-level potentially unstable layer in the WCB inflow region remains potentially unstable until saturation is reached at the lifting condensation level (Schultz and Schumacher, 1999; Sherwood, 2000; Schultz et al., 2000). Once the lifting condensation level is reached, the potential instability can be released leading to the identified rapid convective updrafts ahead of the cold front.

In contrast, the slower slantwise WCB trajectories start their ascent from the boundary layer in a region characterized by weaker potential instability and lower relative humidity (Fig. 5d) and ascend on top of the (cold) frontal region characterized by comparatively large potential stability (Fig. 5b,d).

---

[1] Note that the non-conservation of $\theta_e$ leads to the non-conservation of the equivalent potential vorticity (EPV) along the ascent, which is often considered to be conserved (e.g., Hitchman and Rowe, 2017) for saturated convective motion (neglecting PV modification through the solenoid effect; for details see Cao and Cho, 1995). EPV is defined as PV but with $\theta_e$ replacing $\theta$ ($EPV = \frac{1}{\rho} \boldsymbol{\omega} \cdot \nabla \theta_e$).

### 3.5 Vertical and horizontal PV structure

The PV perspective is useful to understand and trace the effect of convection on the atmospheric circulation. In this section, we investigate the 3D PV structure associated with convective and slantwise WCB ascent and describe the mechanisms that lead
to the differing PV distributions associated with the two types of WCB ascent.

The strong localized diabatic heating during the ascent results in a PV production below and a PV destruction above the strongest ascent phase for both WCB categories (Fig. 5e,f), which is characteristic for WCB ascent (e.g., Wernli and Davies, 1997; Pomroy and Thorpe, 2000; Madonna et al., 2014). In comparison with the more slowly ascending WCB trajectories, in particular the positive PV anomaly formed by convective WCB ascent is much stronger and more localized. The averaged
low-level PV monopole below the convective WCB ascent reaches values of up to 4.5 PVU, while it remains below 1.5 PVU for the slantwise WCB ascent. In the WCB outflow, the PV values decrease to approximately 0.2 PVU for both WCB categories (Fig. 5e,f). Despite the stronger and vertically more extended positive low-level PV anomaly produced by the convective WCB trajectories, both types of WCB trajectories lead to an extended region of low-PV air directly in their outflow in the upper troposphere. In particular the outflow of the convective WCB trajectories is associated with a region of low static stability
($d\theta/dz \leq 2\,K\,km^{-1}$, Fig. 5e, white hatching).

In the following, we examine the PV structure not only along the WCB trajectories, but also in the surroundings of the WCB ascent. Furthermore, we confront the results with the theoretical concept of PV modification and consider the vorticity and static stability structure in the PV anomalies.

#### 3.5.1 Low-level positive PV monopole

Figure 7a shows the PV structure in the lower troposphere below the mean trajectory position 30 minutes after the start of the convective WCB ascent. In this region, below the level of the diabatic heating maximum, the convective WCB ascent leads to the strong positive PV anomaly identified in the along-flow analysis (Fig. 5e). This mesoscale PV monopole with values up to 4 PVU extends horizontally about 30 km around the convective udpraft and is embedded within an environment with a lower-amplitude positive PV anomaly that results from the slower, slantwise WCB ascent (Fig. 7b). In contrast to the
strong mesoscale PV monopole formed by the convective WCB ascent, the increased low-level PV values associated with the slantwise WCB ascent have a lower magnitude of around 1.5 PVU. However, the PV anomaly occurs on a larger spatial scale of up to 100 km, with decreasing amplitude away from the WCB ascent. Hence, due to the stronger and more localized diabatic heating in the convective WCB trajectories (cf. section 3.2), and, as a consequence, stronger PV modification, the mesoscale low-level positive PV produced by convection is superimposed on and embedded in the PV signal resulting from
the larger-scale and slower WCB ascent (Figs. 7a,b).

#### 3.5.2 Upper-level PV dipole

In the middle to upper troposphere a coherent mesoscale horizontal PV dipole forms in the vicinity of the convective WCB trajectories, with a positive PV pole of magnitude 3 PVU to the right of the vertical wind shear vector (which points in the same

direction as the upper-level wind vector) and a negative PV pole of magnitude $-1.5$ PVU to the left of the vertical wind shear
vector (Fig. 7c,e). This PV dipole extends vertically from about 3 km (305 K) to about 9 km (330 K). The maximum amplitude
of the PV dipole occurs at about 315-320 K (Fig. 7e) and coincides with the diabatic heating maximum associated with the
maximum of the formation of snow and graupel (Fig. 4c). Similar to the positive PV monopole at low levels, the upper-level
PV dipole also extends to about 30-40 km around the center of the convective updraft (Fig. 7c).

This distinct mesoscale PV signal emphasizes the coherent signature of the individual convective updrafts that are embedded
within the complex WCB airstream. The robust mesoscale response can only be identified so clearly in the composite analysis.
The large-amplitude, small-scale and, fragmentary PV features that occur in the upper troposphere in the region of embedded
convection on instantaneous PV charts (Figs. 2 and 9d; see also online supplemental material) correspond to such mesoscale
PV dipoles formed by the individual convective updrafts embedded in the WCB.

The formation of the PV dipole above the low-level PV monopole in our composite analysis is directly comparable to the
PV structure of isolated convective updrafts in a sheared environment (cf. Fig. 2 in Chagnon and Gray, 2009) or larger-scale
convective systems as discussed in the introduction. It has, however, not yet explicitly been associated with WCBs identified in
reanalysis data and coarser-scale simulations, where the vertical PV dipole structure dominates (e.g., Wernli and Davies, 1997;
Joos and Wernli, 2012; Madonna et al., 2014).

The composites for the slantwise WCB trajectories reveal the typical PV structure of WCBs (e.g., Wernli and Davies, 1997;
Pomroy and Thorpe, 2000) with a wide region of low-PV air with a magnitude of about 0.5 PVU in the upper-tropospheric
WCB outflow above the low-level positive PV anomaly (Fig. 7b,d,f and 5f). The poleward ascending low-PV air in the WCB
outflow spreads out into the upper tropospheric ridge, potentially leading to its amplification (Grams et al., 2011; Madonna
et al., 2014).

### 3.5.3 Mechanisms leading to the PV structure

To analyse the mechanisms responsible for the formation of these coherent PV anomalies, we consider the material change of
PV in the form of Eq. 4, which emphasizes the contributions from the vertical (first term) and horizontal (second term) vorticity
components and heating gradients as discussed in the introduction (section 1.2).

The formation of the low-level positive PV anomaly is mainly due to the strong vertical heating gradient in an environment
with large cyclonic vertical vorticity (first term in Eq. 4), such that below the diabatic heating maximum PV is increased (e.g.,
Stoelinga, 1996; Wernli and Davies, 1997; Rossa et al., 2000; Joos and Wernli, 2012; Binder et al., 2016). This mechanism
is important for PV modification in both the convective and slantwise WCB trajectories, which ascend near the surface cold
front, i.e., in a region where absolute vertical vorticity $f + \zeta$ is particularly large. Due to the stronger and more localized
diabatic heating in the convective WCB ascent, the vertical heating gradient $\frac{\partial \dot{\theta}}{\partial z}$ is larger and, therefore, the convective WCB
ascent leads to a stronger low-level positive PV anomaly. Furthermore, vortex stretching in the convective updraft additionally
enhances the low-level absolute vertical vorticity. Together, this can lead to a positive feedback mechanism: PV is diabatically
produced in the convective updraft with strong diabatic heating gradients $\frac{\partial \dot{\theta}}{\partial z}$. In consequence, the vertical vorticity is enhanced
supported by vortex stretching. In the following, the diabatic heating gradient acts on amplified vertical vorticity, and thus still

larger PV anomalies are produced. Finally, this results in increased PV production (cf. Joos and Wernli, 2012; Madonna et al., 2014).

The mid- to upper-level convective PV dipole results from the arrangement of horizontal vorticity and the horizontal diabatic heating gradient (second term in Eq. 4, cf. Figs. 1 and 7c). Horizontal vorticity is large ahead of the upper-level trough due to strong vertical wind shear below the upper-level jet. In this case, the vertical wind shear vector of magnitude 2-3 m s$^{-1}$ km$^{-1}$ between 4-12 km height points in the same direction as the upper-level wind vector, i.e., towards the northeast (Fig. 7c). The horizontal vorticity vector $\boldsymbol{\omega_h}$ is rotated 90° anticlockwise relative to the vertical wind shear vector and points towards

the cold air. Moreover, the horizontal heating gradients $\nabla_h \dot{\theta}$ point radially towards the center of the convective updraft, which coincides with the maximum heating from graupel and snow formation (Fig. 4a,e). As outlined in the introduction (section 1.2), this results in PV production to the right of the vertical wind shear vector, where $\nabla_h \dot{\theta} \parallel \boldsymbol{\omega_h}$, and PV destruction to the left of the vertical wind shear vector, where $\nabla_h \dot{\theta} \parallel -\boldsymbol{\omega_h}$ (Fig. 7c). The convectively produced heating gradients and the background vorticity are strong enough to form a region of absolutely negative PV. These findings from the convective WCB composites

agree with the theoretical considerations in the introduction.

    The horizontal diabatic heating gradients for the slantwise WCB ascent are weaker because the diabatic heating is (i) less intense and (ii) spatially more uniform due to a larger-scale gradual ascent (cf. Fig. 4d,f). Thus, the vertical component of the PV equation is relatively more important than the horizontal component for the large-scale slantwise WCB ascent, and continuous heating along the ascent leads to PV reduction above and a transport of low-PV air to the upper troposphere by

395 the trajectories passing through this low-PV region. However, PV values remain positive because PV cannot change sign if the first term in Eq. 4 dominates (Harvey et al., 2020). We conclude that for the slantwise WCB ascent the vertical component of the PV equation is most relevant, while for embedded convection with localized and intense heating gradients the horizontal components of the PV equation are essential to explain the resulting upper-level PV dipole structure, which includes negative PV values.

### 400   3.5.4   Partitioning of PV anomalies in vorticity and static stability

Negative PV implies either hydrostatic, inertial or symmetric instablility (e.g., Hoskins, 2015). In the following, we analyse the partitioning of the PV anomalies in vorticity and static stability and discuss its implication for the expected lifetime of these anomalies. The PV dipole is associated with a dipole of absolute vertical vorticity $f + \zeta$ with similar magnitude in both poles (Fig. 8b), and thus, can be understood as the effect of tilting of horizontal vorticity by the convective updraft (cf.

Chagnon and Gray, 2009). Moreover, strong heating in the convective updraft leads to increased static stability inside the updraft ($d\theta/dz = 3\text{-}5 \, \text{K km}^{-1}$; not shown) and a shallow layer of low static stability ($d\theta/dz \leq 2 \, \text{K km}^{-1}$, Figs. 5e and 7e) above. The static stability is relatively uniform across both poles. Hence, the PV dipole's horizontal structure is predominantly determined by vorticity and not static stability. This is consistent with Chagnon and Gray (2009), who also found that in convective updrafts the so-called 'latent vorticity' is quickly converted to horizontal dipoles of vertical vorticity that determine

the PV dipole structure.

In section 4.2 we will see that the negative PV pole, produced by convective WCB trajectories, persists for several hours. This is consistent with adjustment timescales of several hours (e.g., Thompson et al., 2018) in inertially unstable regions where $f + \zeta < 0$. A convectively unstable atmosphere ($\mathrm{d}\theta/\mathrm{d}z < 0$), in contrast, would adjust to stability on a timescale of less than 1 h (Schultz and Schumacher, 1999).

The static stability reduction above the slantwise WCB is weaker compared to the convective WCB, but the slantwise WCB leads to a reduced static stability over a larger region (Fig. 7f). In the outflow of the slantwise WCB relative vorticity is weakly negative and absolute vertical vorticity is weakly positive (not shown).

     The low-level positive PV anomalies for convective and slantwise WCB ascent are associated with large cyclonic vertical vorticity $f + \zeta$, whereby the convective WCB has higher values, exceeding $6 \cdot 10^{-4}\,\mathrm{s}^{-1}$ (Fig. 8a), compared to the slantwise
WCB with values of $(3\text{-}4) \cdot 10^{-4}\,\mathrm{s}^{-1}$ (not shown).

### 3.6    Flow anomalies induced by PV anomalies

In agreement with idealized PV inversions (e.g., Hoskins et al., 1985; Hoskins, 2015), where a positive PV anomaly induces a cyclonic flow anomaly, and a negative PV anomaly induces an anticyclonic circulation anomaly, respectively, the convectively produced PV dipoles are associated with coherent horizontal wind anomalies, calculated as the deviation of the current wind
vectors from the 2-h centered mean wind vectors. The low-level positive PV monopole is accompanied by a cyclonic circulation anomaly with about $4\,\mathrm{m\,s}^{-1}$ higher wind speeds southeast of the convective updraft and smaller values to the northeast (Fig. 7a,e). Despite this relatively strong local wind anomaly, its radius of impact is limited and the effect of the PV anomaly substantially decays beyond $40\,\mathrm{km}$ from the updraft. As hypothesized by Raymond and Jiang (1990), the relatively long-lived low-level positive PV anomalies interact with the background shear, and thus could trigger new convective cells through the
formation of local convergence lines on the downshear side. In this way convective activity could be maintained.

     The positive PV monopole from the slantwise WCB trajectories also induces a cyclonic low-level circulation anomaly (Fig. 7b). Yet, it hardly exceeds $1\,\mathrm{m\,s}^{-1}$ and occurs on a larger spatial scale, in agreement with the comparatively weaker and larger-scale positive PV anomaly. The initial slantwise WCB ascent occurs directly behind the pronounced low-level jet ahead of the cold front (Fig. 7b,f). This jet, which exceeds $24\,\mathrm{m\,s}^{-1}$, is very slightly accelerated by the diabatically produced positive
PV and the associated cyclonic circulation anomaly ahead of the WCB ascent region. This pattern agrees with the synoptic situation of early WCB studies (e.g., Fig. 5 in Wernli, 1997), where the ascent region of the slantwise WCB is located ahead of the upper-level jet and behind the low-level jet.

     The convectively produced upper-level PV dipole is associated with a cyclonic and anticyclonic circulation anomaly around the positive and negative PV poles, respectively (Fig. 7c,e). The superposition of these two flows leads to a deceleration of the
flow in the center of the PV dipoles and potentially stabilizes the convective cloud against rapid advection with the upper-level flow (cf. Oertel, 2019, Chapter 7). At 320 K, this induced wind anomaly reaches almost $3\,\mathrm{m\,s}^{-1}$ close to the convective updraft.

     The weaker negative upper-level PV anomaly of the more slowly ascending WCB trajectories goes along with a widespread weak anticylonic circulation anomaly (Fig. 7d) with a maximum anticyclonic wind speed anomaly of less than $0.5\,\mathrm{m\,s}^{-1}$

northwest of the WCB ascent. The more slowly ascending WCB trajectories arrive in the upper troposphere in the vicinity of the tropopause.

In summary, both WCB categories are associated with a cyclonic low-level circulation anomaly induced by the low-level positive PV anomaly. The wind anomaly in the convective case is stronger but the extent is limited, while the wind anomaly in the slantwise WCB category is substantially weaker but extends to a larger region. In the upper troposphere two different circulation anomalies establish. The slantwise WCB ascent induces a widespread and comparatively weak anticyclonic circulation anomaly. In contrast, the anticyclonic and cyclonic circulation anomalies induced by the convectively generated PV dipole occur on a smaller scale and lead to a deceleration of the flow in the center of the convective updraft.

## 4   PV anomalies on a larger scale and relevance for large-scale dynamics

Section 3 showed that the individual convective updrafts are associated with mesoscale upper-level PV dipoles that are, however, too small-scale to directly interact with the synoptic-scale balanced flow (cf. Shutts, 2017). Section 4.1 illustrates that these fragmentary PV anomalies can indeed aggregate to larger-scale PV dipole bands. Thereafter, we assess the potential for the interaction of convectively produced PV dipole bands with the synoptic-scale flow. For this, we coarse-grain the PV field with a 60-km smoothing radius (cf. Shutts, 2017; Clarke et al., 2019). More specifically, we project the original 2 km PV field to a coarser grid using a spatial moving average over $60 \times 60\,\text{km}^2$ directly on selected isentropes.

### 4.1   An illustrative example of WCB-embedded convection

Figure 9 shows an instantaneous example of convection embedded in the large-scale WCB cloud structure at 09 UTC 23 Sep 2016 and serves to illustrate the typical properties and characteristics deduced from the composite analysis in a real synoptic context. Based on this example we will discuss the lifetime of the convectively generated PV dipoles and their potential for the interaction with the larger-scale flow (section 4.2).

The large-scale WCB cloud band (Fig. 9a) is heterogeneously structured with a strong and localized production of graupel, snow and rain (Figs. 9b and 10a) caused by enhanced updrafts from embedded convection. Embedded convection, identified by rapidly ascending WCB trajectories (Fig. 9c, white contours), is predominantly located in the baroclinic region ahead of the upper-level trough (Fig. 9a and 9c, white contours). Indeed, the online trajectories' ability to identify the convective ascent is confirmed by the vertical cross-section through an embedded convective updraft (Fig. 10a): A substantial amount of hydrometeors is locally produced inside the convective updraft. This spatial coincidence of pronounced hydrometeor production within the updrafts (compare Figs. 9b,c and Fig. 10a) agrees with the results from the composite analysis (Fig. 4a,c).

Although the upper-level PV associated with the convective updrafts is fragmented into many small-scale features (Fig. 9d), the PV structure identified in the composite analysis (cf. Fig. 7e) is clearly discernible in a vertical cross-section through the convective updraft (Fig. 10b): in the lower troposphere a strong positive PV monopole forms, which is replaced by a horizontal PV dipole centered around the convective updraft in the mid- to upper troposphere. The PV dipole extends from around 4 km to 11 km height and the PV values exceed $\pm 10\,\text{PVU}$, leading to large horizontal PV gradients of up to $1\,\text{PVU}\,\text{km}^{-1}$.

In agreement with the composite analysis, the static stability at the height of maximum diabatic heating at around 320 K is increased (not shown), while above the heating maximum a lens of low static stability forms across the negative and positive PV poles (Fig. 10b, white contour and hatching). Thus, the mesoscale PV dipole pattern with an extent of approximately 100 km across both poles originates predominantly from the spatial variability of vertical vorticity, in agreement with the composite analysis (Fig. 8b and section 3.5).

The composite analysis (section 3.4) revealed that the convective WCB trajectories ascend in a region characterized by substantially higher $\theta_e$ that forms a narrow elongated tongue of warm and moist air ahead of the cold front. Figure 9e emphasizes the strong heterogeneity of $\theta_e$ in the warm sector and confirms the spatial coincidence of rapid WCB ascent (Fig. 9e, white contours) and the localized narrow structures of high $\theta_e$ air. This underlines the role of the mesoscale temperature and humidity gradients for triggering convection that is directly embedded within the large-scale slantwise WCB ascent.

In the lower troposphere an elongated zone of horizontal wind convergence concurs with the convective updraft region (Fig. 9f). This low-level convergence line coincides with a band of increased low-level PV (Fig. 9f, contour lines), which is generated and potentially further enhanced by continuous convective ascent (cf. section 3.5).

On the larger-scale (after a coarse-graining of the PV field to a $60 \times 60\,\text{km}^2$ grid), the convective region forms a meridionally elongated and narrow upper-level PV dipole band with an extension of 100 km in the across-front and more than 400 km in the along-front direction, respectively (Fig. 9c). The orientation of this dipole band is aligned with the vertical wind shear vector and the elongated narrow band of convective activity ahead of the cold front and the upper-level trough, and is directed to the northeast. Hence, despite the small-scale noise in the PV field formed by the individual convective updrafts (Fig. 9d), the PV anomalies spatially aggregate to a coherent and robust PV structure on the larger scale (Fig. 9c). The formation of this elongated PV dipole band parallel to the convective region is consistent with the composite analysis (Fig. 7c) and theoretical considerations (cf. Eq. 4 and Fig. 1). The formation of these PV dipole bands on either side of elongated convective ascent regions can be observed at various times ahead of the upper-level trough in this WCB case study (not shown).

A larger-scale circulation anomaly establishes around the coarse-grained PV dipole band with cyclonic and anticyclonic wind anomalies around the positive and negative poles, respectively (Fig. 9c). The wind anomalies scale with the amplitude of the PV anomalies and are particularly strong in the region where the coarse-grained PV anomalies exceed $\pm 2\,\text{PVU}$. Note that the wind anomalies are not coarse-grained, emphasizing that the organized mesoscale PV features aggregate to coherent PV anomalies, which are directly associated with a dynamical response on the larger scale.

The associated circulation anomaly leads to an increase of the wind speed to the northwest and southeast of the negative and positive pole, respectively, and to a deceleration of the flow in the PV dipole's center (Fig. 9c,d). Although the far-field effect of the PV anomalies reaches beyond the region where PV is directly modified, the upper-level waveguide is located more than 250 km to the northwest of the convective band and, therefore, at this particular time when convection forms the PV dipole, the circulation anomaly does not yet directly influence the upper-level waveguide.

In summary, this instantaneous, Eulerian analysis of WCB-embedded convection agrees with the composite analysis in section 3 and illustrates the typical properties of WCB-embedded convection: strong and localized diabatic heating inside a

510 dense and precipitating cloud leads to the formation of mesoscale upper-level PV dipoles that are associated with a coherent larger-scale circulation anomaly.

## 4.2 Temporal evolution of the convectively generated upper-level PV dipole

At the time when the PV dipole bands are formed, the associated circulation anomalies do not reach the upper-level waveguide and jet stream in 250 km distance as they are confined to within about 100 km around the PV dipole band (Figs. 9c and 11a).
Yet, the temporal evolution of upper-level PV reveals that, even after the convective updrafts cease, the negative PV band persists (Fig. 11a-d): The evolution of the negative PV feature under consideration (blue contour in Fig. 11 with pink shading) shows that during its relatively long lifetime of several hours, the negative PV pole is advected by the upper-level wind to the north and toward the dynamical tropopause. The negative PV appears to be approximately conserved and, a couple of hours after the convective updraft ceases, the negative PV band has approached the upper-level jet (distance now only about 150 km)
and the adjacent wind speed at the jet is increased by approximately $5 \, \mathrm{m \, s^{-1}}$ to more than $60 \, \mathrm{m \, s^{-1}}$ (Fig. 11c).

To confirm the advection of the negative-PV air by the upper-level flow, forward trajectories are started inside the negative PV anomaly at the time when the PV dipole band is formed by the embedded convection (Fig. 11a, pink shading; for more details about the trajectory computation see below): The positions of these forward trajectories (Fig. 11a-d, pink shading) mostly coincide with a negative PV region, in particular within the first 3-6 h.
After 9 hours, the negative PV pole, distorted in shape due to the influence of strong horizontal wind shear in the jet region, closely approaches the waveguide and thereby strongly increases the isentropic PV gradient near the tropopause (Fig. 11d-e). The strong anticyclonic circulation anomaly associated with the negative PV anomaly accelerates the jet and forms a local jet streak (Figs. 11d,e) that is maintained for 2-3 h. This local jet intensification occurs although during advection of the negative PV anomaly, its magnitude decreases (Fig. 11f) compared to the initial strength immediately after its formation (Fig. 10b).
The area covered by negative PV values, on the other hand, increases on the considered isentrope (Figs. 11a-d, blue contours). This negative PV anomaly in the ridge ahead of the upper-level trough appears to amplify the amplitude of the pre-existing PV pattern at the waveguide.

With increasing time after the initial PV dipole formation, the negative PV pole overtakes the positive PV pole (Fig. 11a-d, orange and red contours) due to strong horizontal wind shear, i.e., decreasing upper-level wind with increasing distance to the
535 jet. In contrast to the negative PV pole that is advected towards the waveguide, the positive PV anomaly remains in the ridge and away from the waveguide.

## 4.3 Analysis of trajectories initialized at 09 UTC 23 Sep

The origin and fate of the negative-PV air is analysed in more detail using offline trajectories with 15-minute temporal resolution. For this, forward and backward trajectories are computed from a region of convectively produced negative PV (Fig. 11a,
pink shading). We only select trajectories with negative PV values between 315 K and 325 K that are located within a larger region of spatially averaged negative PV at 09 UTC 23 Sep (Figs. 11a and 12a).

For the analysis of trajectories starting in the upper troposphere, offline (in contrast to online) trajectories have to be considered because the online trajectories are only started in the lower troposphere to obtain a large number of strongly ascending trajectories. Due to computational costs (memory allocation) the number of online trajectories is limited. As a consequence, online trajectories arriving in the upper toposphere have all performed a deep ascent from lower levels. However, these strongly ascending trajectories do not primarily experience the strongest PV modification (see below). Hence, the amount of available online trajectories in the target region (i.e., the region with negative PV in the upper troposphere) is too small. Finally, the online trajectories can only be computed forward, and not backward.

With the backward trajectories, however, we can infer when, relative to 09 UTC 23 Sep (which we will refer to as $t=0$ h), the negative-PV air masses acquired their negative PV and where they originate from (Fig. 12). Six hours before, less than 10% of these trajectories have negative PV values, while 3 h before, this percentage amounts to 30% (Fig. 12b, black curve). After $t=-1$ h more than 40% of trajectories additionally acquire a negative PV value through the *remote effect* of localized heating as they pass through a convectively influenced region, i.e., a region to the left of the convective updraft region and the vertical wind shear vector, where PV is reduced (cf. Fig. 1). Thus, the majority of trajectories acquires their negative PV just within the last hour while passing a convectively influenced region, predominantly located west of the convective updraft region.

From all trajectories with negative PV at $t=0$ h, almost 70% were previously advected quasi-isentropically with the upper-level flow and a smaller percentage of 30% ascended from the lower to the mid-troposphere (Fig. 12b, grey curves and Fig. 12c), few of them inside a convective region. The percentage of upper-level trajectories with negative PV advected into the negative-PV region increases with time (Fig. 12b, solid blue curve) because upstream convection produced negative upper-level PV 1-3 h earlier (not shown). This air mass is then advected and contributes to the larger-scale region of negative PV at $t=0$ h. The strongest increase of the percentage of upper-level trajectories with a negative PV value occurs at $t=-1$ h (Fig. 12b, solid blue curve), when strong convection sets in and reduces the PV of the trajectories to the left of the convective updraft. Trajectories that ascend from lower levels (Fig. 12c) and contribute to the larger-scale region of negative PV at $t=0$ h have PV values of $\pm 10$ PVU (Fig. 12a). However, the percentage of trajectories with negative PV ascending from the lower troposphere is small and does not exceed 10% (Fig. 12b, dashed blue curve).

These results suggest that the larger-scale region of upper-level negative PV consists of air masses with different origins: (i) a large percentage of trajectories (approximately 30%) with negative PV is formed by upstream convection approximately 3 h earlier and is advected quasi-isentropically by the upper-level flow; (ii) a substantial fraction of air parcels (approximately 40%) originating from the upper troposphere acquires negative PV values only within the last 1 h through local convective influence as they pass the left side of a convective updraft; (iii) only a small fraction of trajectories contributing to the upper-level negative PV at $t=0$ h originates from the lower to mid-troposphere with positive PV (approximately 10% of the trajectories are located in the lower troposphere and have positive PV values at $t=-3$ h), whereby these rising trajectories acquire a negative PV value during their ascent or after arrival in the upper troposphere when they pass through a convectively influenced region; (iv) the smallest fraction of less than 10% contains trajectories that ascend from lower levels with already negative PV 3 h prior to their ascent.

Although there is some uncertainty in the calculation of the offline trajectories with 15-minutes temporal resolution, these results suggest that air masses that acquire negative PV originate from a region in close proximity to convection but not necessarily from inside the convective updraft. Moreover, this indicates that the maintenance of convective ascent for at least a few hours helps to generate a larger region of negative PV through advection of negative-PV air from upstream convection.

Whereas the backward trajectories allowed analysing the PV history of the air parcels, the forward trajectories consider the future evolution of their PV. Within the first 3-6 h, the forward trajectories remain mostly within a negative PV region (Fig. 11a-d, pink shading) and the majority of trajectories keep their negative PV values (Fig. 12a). This emphasizes the persistence of the negative PV and supports the previous finding that the negative PV region is advected by the upper-level flow (Fig. 11). After 6 h, the trajectories spread out spatially and cover a larger region due to the strong horizontal and vertical wind shear, but 57% of the trajectories still have negative PV. The negative PV is retained for a relatively long time, and even after more than 12 h more than 50% of the trajectories still have negative PV (Fig. 12a) and contribute to some extent to a widespread larger-scale negative PV anomaly (Fig. 11d-e, pink shading), which induces an anticyclonic circulation anomaly in proximity to the tropopause.

Together, these results suggest that air masses in the vicinity of convective updrafts experience strong PV destruction, which contributes to a larger region of negative PV with a coherent anticyclonic circulation anomaly. The convectively generated negative PV is characterized by a relatively long lifetime and is advected northward towards the upper-level jet, where it interacts with the upper-level waveguide, strengthens the isentropic PV gradient and is associated with the formation of a jet streak.

## 5   Discussion and open question

In the following we discuss our results, remaining open questions, and limitations of this study. In this WCB case study, embedded convection was frequently observed and consistently associated with elongated upper-level PV dipole bands, whereby the negative PV pole was advected poleward towards the jet where it interacted with the waveguide. However, the analysis of only one case study limits the generality of the key results, i.e., the frequent occurrence of convectively produced PV dipole bands in WCBs and their influence on the upper-level jet. Due to the wide range of synoptic situations that can be associated with WCB ascent, we expect a large case-to-case variability of the resulting PV signal. For example, a case study of a WCB associated with an upper-level PV cut-off in October 2016 in the North Atlantic region generated weaker and less consistent PV dipoles (Oertel, 2019, Chapter 6).

Despite the large variability of WCB ascent behavior, we hypothesize that the WCB ascent regions are generally favorable environments for the production of PV dipole bands and the occurrence of negative PV in proximity to the tropopause (cf. Harvey et al., 2020), as they frequently ascend in the baroclinic region of extratropical cyclones characterized by substantial vertical wind shear. While the occurrence of negative PV in ridges of mid-latitude cyclones in proximity to the tropopause has already been reported from model simulations (e.g., Pomroy and Thorpe, 2000; Grams et al., 2011; Chagnon et al., 2013; Rowe and Hitchman, 2016; Harvey et al., 2020) and observations (Harvey et al., 2020), it has not yet been explicitly associated

with embedded convection in WCBs. The extension of this analysis to further WCB case studies – including the investigation of embedded convection, its PV signature and its dynamical relevance – would shed light on the generality of our key results. In particular, the identified impact of embedded convection on the dynamics and precipitation pattern requires a climatological quantification of the frequency of convective versus slantwise WCB ascent. However, the investigation of numerous WCB case studies in convection-permitting models is computationally expensive, also because of the large domain covered by the WCB and the need for online trajectories.

A study of an extratropical cyclone by Chagnon et al. (2013) suggested that diabatic processes do not necessarily alter the PV directly at the tropopause, but rather in its environment, resulting in the steepening of the PV gradient across the tropopause. They showed that negative PV had been diabatically modified within the WCB outflow region at the equatorward side of the upper-level waveguide, and subsequently stretched out in the ridge through non-linear advection, similarly to the behavior of the convectively generated negative PV during poleward advection into the ridge in this case study (section 4.2, Fig. 11). In their case study, the negative PV, resulting from parameterized convection, large-scale microphysical processes and boundary layer processes, was in phase with the meridional PV gradient. This resulted in the amplification of the wave, and thus was associated with faster Rossby wave growth rates and enhanced westward propagation relative to the flow. Similarly, the negative PV bands formed by WCB-embedded convection close to the tropopause are in phase with the meridional PV gradient, and thus also enhance the wave amplitude and steepen the PV gradient. Hence, the advection of the convectively generated negative PV pole towards the waveguide further enhances the horizontal PV gradient at the tropopause, in addition to the increase of the tropopause sharpness by the slantwise WCB ascent as reported by Chagnon et al. (2013) and Chagnon and Gray (2015). This approximation of negative PV and the waveguide also locally accelerates the jet and potentially influences the propagation of Rossby waves (Harvey et al., 2016), both highly relevant aspects for numerical weather prediction.

During its poleward advection towards the tropopause, the negative PV pole wraps anticyclonically due to the strong wind shear near the jet and expands in the ridge (cf. Chagnon et al., 2013). This expansion increases the influence of PV on the induced velocity, as larger-scale PV anomalies more strongly affect the velocity field (Hoskins et al., 1985). Moreover, this stresses that, although embedded convection locally strongly impacts the precipitation, cloud structure, and the mesoscale flow during the convective updraft, the impact on the large-scale dynamics occurs mostly downstream and several hours after the convective updraft ceases.

As discussed in the previous paragraph, the propagation of convectively generated bands of negative PV towards the waveguide is dynamically relevant due its impact on the tropopause sharpness, on the upper-level jet and potentially on the propagation speed of Rossby waves. This interaction of the negative PV pole with the upper-level waveguide can be observed several times during the life cycle of *Vladiana* (e.g., Fig. 11). Based on the analysis of these examples, Fig. 13 schematically illustrates the different stages of convectively generated negative PV embedded in the WCB ascent region from its initial formation to its slow decay at the anticyclonic shear side of the upper-level ridge.

Initially, the aggregation of embedded convection parallel to the upper-level jet forms a coherent larger-scale elongated PV dipole band parallel to the upper-level jet (Fig. 13a). After its initial formation, the negative PV band overtakes the positive

PV band due to the horizontal wind shear near the jet, and gets distorted in shape (Fig. 13b). This results in a re-orientation of the PV dipole bands, such that the distorted, almost circular negative and positive PV features are meridionally aligned, with the negative PV to the north and the positive PV pole to the south. At this stage, the PV dipoles induce a circulation anomaly towards the west that facilitates the westward propagation of the negative PV relative to the approaching waveguide, which is the prerequisite for the direct interaction between the upper-level jet and the negative PV feature. As the propagation of the negative PV band towards the waveguide is observed several times in this case study (shown exemplarily for one such event in section 4.2), we hypothesize that the induced wind field at this stage is a coherent feature that supports the advection of the negative-PV air westward relative to the approaching trough. Once the negative PV feature is located in close proximity to the waveguide in the northern part of the ridge, it is distorted and stretches out (Fig. 13c), and finally forms a narrow band of negative PV directly at the anticyclonic shear side of the jet (Fig. 13d), where the magnitude of the negative PV slowly decays. Figure 13 also illustrates the acceleration of the upper-level jet (Figure 13b) and the formation of a jet streak (Figure 13c,d), when the negative PV pole approaches the upper-level jet (cf. Harvey et al., 2020).

Currently, global forecast models (e.g., IFS, MetUM, GFS) are run with a coarser resolution unable to explicitly resolve convection. This leads to the question if these schemes are able to trigger embedded convection at the right place and time and with the correct amplitude, and if the PV dipoles generated by localized heating from the convection parameterization scheme have the correct magnitude. For instance, previous studies showed that the PV anomaly's amplitude generated by the convection scheme is weaker compared to the amplitude in high-resolution convection-permitting simulations (Done et al., 2006; Clarke et al., 2019), which, in the case of a large mesoscale convective system, subsequently influenced the downstream flow evolution and decreased the forecast quality (Clarke et al., 2019). To explicitly simulate the mesoscale upper-level PV dipoles, a horizontal grid spacing of approximately 2 km would be required to resolve the individual convective updrafts.

The organized mesoscale PV features in the convective ascent region on the original 2 km grid can aggregate to coherent PV anomalies on the large-scale, which are directly associated with a dynamical response on the large-scale. This agrees with the electrostatics analogy of PV (Bishop and Thorpe, 1994), which suggests that locally confined PV 'charges' each induce a certain far-field effect on the flow and the superposition can be attributed to the spatially integrated PV anomaly. This implies that the linear superposition principle is applicable to PV, which denotes that the effects of individual PV anomalies on the flow field are additive (Bishop and Thorpe, 1994; Birkett and Thorpe, 1997). Although the linear superposition is only exactly valid for quasi-geostrophic PV, Thorpe and Bishop (1995) and Birkett and Thorpe (1997) suggested that the non-linear contributions for Ertel-PV decrease with distance and with decreasing amplitude of the PV anomaly, and can therefore likely be neglected. This study supports their conclusion because the convective PV anomalies aggregate to coherent larger-scale PV anomalies with a distinct effect on the flow field.

While embedded convection is directly relevant for numerical weather prediction due to its potential impact on surface precipitation, mesoscale winds, jet speed and Rossby wave propagation, the relevance of correctly representing embedded convection in climate simulations is less obvious. Nevertheless, the misrepresentation of embedded convection in WCBs could underestimate the projected surface precipitation extremes associated with extratropical cyclones. Moreover, the large-scale

cloud band of WCBs is known to modify the radiative balance (Joos, 2019), which is still a highly uncertain process in climate simulations (e.g., Boucher et al., 2013; Vial et al., 2013; Bony et al., 2015; Caldwell et al., 2016). A systematic underestimation of embedded convection and its influence on the cloud structure could potentially influence the radiative balance, because, on the one hand, convection results in a substantially denser cloud band. On the other hand, convectively generated liquid-origin cirrus clouds (cf. Krämer et al., 2016; Luebke et al., 2016; Wernli et al., 2016) spread out in the WCB outflow and travel poleward. The formation pathway of cirrus clouds leads to different microphysical and macrophysical properties (e.g., Krämer et al., 2016; Luebke et al., 2016), which can influence the cirrus cloud radiative forcing (Zhang et al., 1999; Joos et al., 2014).

Finally, we showed that the formation of the PV dipole through embedded convection is a relatively fast processes, while the decay of the (dynamically unstable) negative PV pole, with a lifetime of several hours, is comparatively slow. It is not yet clear which non-conservative processes, e.g., turbulence, microphysics or radiation, lead to the destruction of the negative PV in the ridge, i.e., the production of PV in regions of negative PV, and if other non-conservative processes additionally decrease PV resulting in the maintenance of the negative PV feature. A novel diagnostic to analyse the PV tendencies from each parameterization scheme available for the ECMWF's Integrated Forecasting System (Spreitzer et al., 2019; Attinger et al., 2019) would enable a detailed and systematic analysis of the processes that govern the destruction of negative PV. It would also potentially shed more light on the approximate lifetime and properties of the negative PV in the upper-level ridge. In addition, idealised high-resolution simulations might provide new insights into the integral lifecycle of negative PV.

# 6 Conclusions

We analysed the influence of embedded convection in the WCB of the North Atlantic cyclone *Vladiana* in Sep 2016 on the precipitation and cloud structure, and on the local mesoscale dynamics and larger-scale circulation features. For this, two categories of online WCB trajectories – very rapidly ascending "convective" WCB trajectories and more slowly ascending "slantwise" WCB trajectories – were identified in a convection-permitting COSMO simulation, and their impact was investigated in a composite analysis and, in more detail for a representative example. As expected from previous studies, the slantwise WCB ascent influences the large-scale precipitation pattern and the hydrometeor and PV distributions in a wide region. The specific signatures of embedded convection are superimposed on these larger-scale patterns from the slantwise WCB ascent. Based on the composite analysis and the investigation of a representative example of embedded convection in the WCB we conclude the following:

Convective WCB ascent leads to substantially stronger surface precipitation during the ascent with on average more than twice the intensity compared to the slantwise WCB ascent (cf. Oertel et al., 2019). The occurrence of embedded convection is therefore relevant for the mesoscale precipitation pattern in cyclones and for extreme surface precipitation associated with WCBs (Pfahl et al., 2014). Moreover, the strong convective updrafts allow for graupel formation, which is absent in the slantwise WCB ascent. Likewise, convective WCB ascent leads to a denser cloud structure in the convective updraft that is embedded within a larger-scale and less dense cloud band (cf. section 1 question 1).

The convective WCB trajectories originate from a region with higher $\theta_e$ and increased potential instability ($\mathrm{d}\theta_e/\mathrm{d}z$) at approximately 900 hPa compared to their environment, which allows for the rapid quasi-vertical convective ascent through the whole troposphere. Although convective WCB ascent is embedded within the larger-scale region of slantwise WCB ascent, mesoscale $\theta_e$ variability and strong localized $\theta_e$ gradients in the warm sector give rise to the higher initial $\theta_e$ of the convective compared to the slantwise WCB trajectories, which subsequently leads to more intense cloud diabatic processes and stronger cross-isentropic ascent (cf. section 1 questions 2 and 3).

The strong horizontal diabatic heating gradients associated with the localised convective WCB ascent results in the formation of mid- to upper-level horizontal PV dipoles on both sides of the convective updraft, where the negative pole (with a diameter of approximately 30 km) occurs to the left of the vertical wind shear vector (cf. Harvey et al., 2020), i.e., relatively close to the upper-level jet (cf. section 1 question 4). The level of maximum amplitude of the PV dipole at around 315-320 K coincides approximately with the diabatic heating maximum associated with the maximum of snow and graupel formation. These convectively produced PV dipoles are predominantly anomalies of absolute vertical vorticity. Static stability inside the convective updraft and across both poles is slightly increased compared to the environment due to strong latent heat release in the updraft. Above the updraft, at about 10 km height, a shallow region of low static stability located inside a cirrus cloud spreads out horizontally.

The low-level and upper-level mesoscale PV features of the convective WCB trajectories are associated with a coherent circulation anomaly, indicating that PV invertibility is qualitatively also valid at this scale. The positive low-level PV monopole below the convective updraft center induces a cyclonic wind anomaly resulting in a local increase of the low-level wind velocity to the right of the vertical wind shear vector, and a decrease to the left, respectively. The upper-level PV dipole is associated with cyclonic and anticyclonic circulation anomalies around the positive and negative poles, respectively (cf. section 1 question 5). The superposition of the circulation anomalies induced by these PV anomalies decelerates the flow near the convective updraft and potentially stabilizes the convectively generated PV dipole against the background flow (cf. Oertel, 2019, Chapter 7).

On the larger scale, the individual mesoscale PV dipoles can, in certain synoptic situations as in this case, aggregate to elongated PV dipole bands downstream of the convective updraft region, which are associated with coherent larger-scale circulation anomalies. The negative PV pole is characterized by a relatively long lifetime of several hours. During its lifetime it is advected close to the upper-level waveguide, where it strengthens the isentropic PV gradient. Its anticyclonic circulation anomaly can contribute to the formation of a jet streak more than 1300 km downstream of the convective updraft and the initial formation of the PV dipole. This suggests that the mesoscale PV anomalies produced by embedded convection upstream can interact with the synoptic-scale circulation, the upper-level jet, and potentially influence the downstream flow evolution, and are thus dynamically relevant (cf. section 1 question 6).

Finally, our mesoscale view on the WCB reveals its heterogeneity. The distinction between slantwise and convective WCB trajectories emphasizes the significantly different impacts on the cloud and precipitation structure and the local mesoscale and larger-scale dynamics, whereby the convective WCB trajectories are accountable for distinct properties, such as the formation of graupel and an upper-level PV dipole, which are absent for purely slantwise ascending WCB trajectories.

745 *Data availability.* All data are available from the authors upon request.

*Video supplement.* Supplemental information related to this paper is available from the ETH research collection at https://doi.org/10.3929/ethz-b-000392157.

*Author contributions.* AO performed the simulation and the data analysis, and prepared a first version of the paper. All authors continuously discussed the results and contributed to the final manuscript.

*Competing interests.* The authors declare that they have no conflict of interest.

*Acknowledgements.* AO and MB acknowledge funding by the Swiss National Science Foundation (Project 165941). The COSMO simulations were performed at the Swiss National Supercomputing Centre (CSCS), as part of the project sm08 (2017–2019). We appreciate the data from EUMETSAT for the visualization of the WCB cloud band and would like to thank Annette K. Miltenberger for her support with the COSMO online trajectory module. Finally, we greatly appreciate the constructive and detailed feedback by Florian Pantillon and Jeffrey 755 Chagnon which helped to further improve the quality of this article.

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

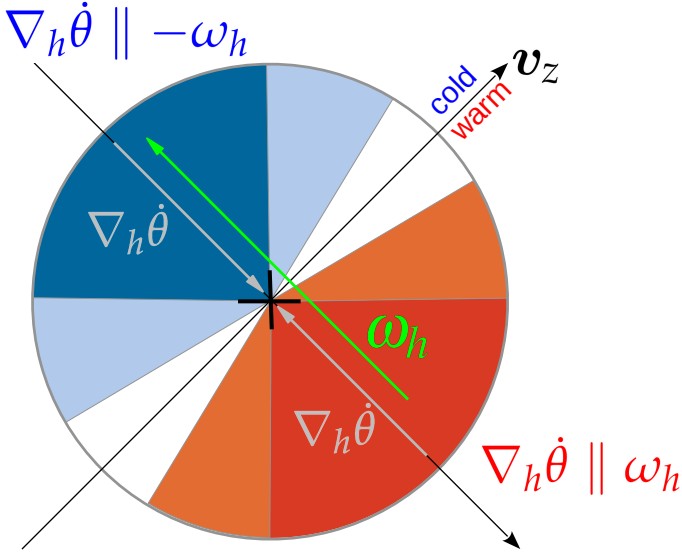

**Figure 1.** Conceptual model of upper-level PV modification by a convective updraft (+) that is located in an environment with background horizontal vorticity. Shown are vertical wind shear vector ($v_z$, black) with the cold air on the left, the horizontal vorticity vector ($\omega_h$, green), the diabatic heating gradients pointing radially toward the convective updraft ($\nabla_h\dot{\theta}$, grey), and the regions where PV is destroyed (blue shading) and where PV is generated (red shading), because $\nabla_h\dot{\theta} \parallel -\omega_h$ and $\nabla_h\dot{\theta} \parallel \omega_h$, respectively [cf. Eq. 4 and Schemm (2013)]. The intensity of the blue and red shading schematically represent the degree of PV change. See text for details. Note that the orientation of the vertical wind shear vector and the PV dipole schematically represents the synoptic situation shown in Fig. 7c.

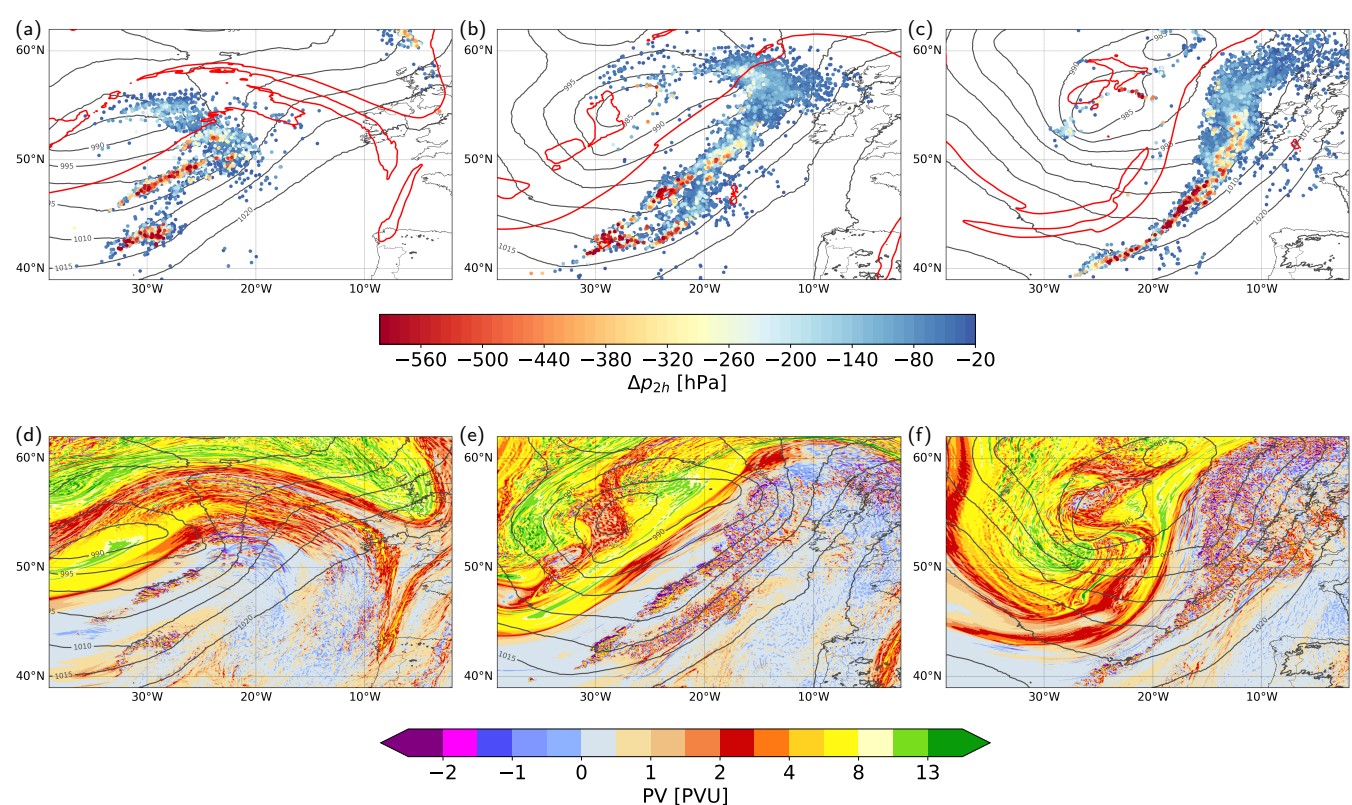

**Figure 2.** Overview of the WCB ascent for cyclone *Vladiana*: **(a-c)** locations of WCB air parcel ascent (circles; colors indicate centered 2-h pressure change ($\Delta p_{2h}$) along all ascending WCB trajectories, in hPa), SLP (grey contours, in hPa) and 2-PVU at 320 K (red line) for (from **a** to **c**) 22 UTC 22 Sep, 10 UTC 23 Sep, and 22 UTC 23 Sep. **(d-f)** PV at 320 K and SLP (grey contours, in hPa) for the same times as **(a-c)**.

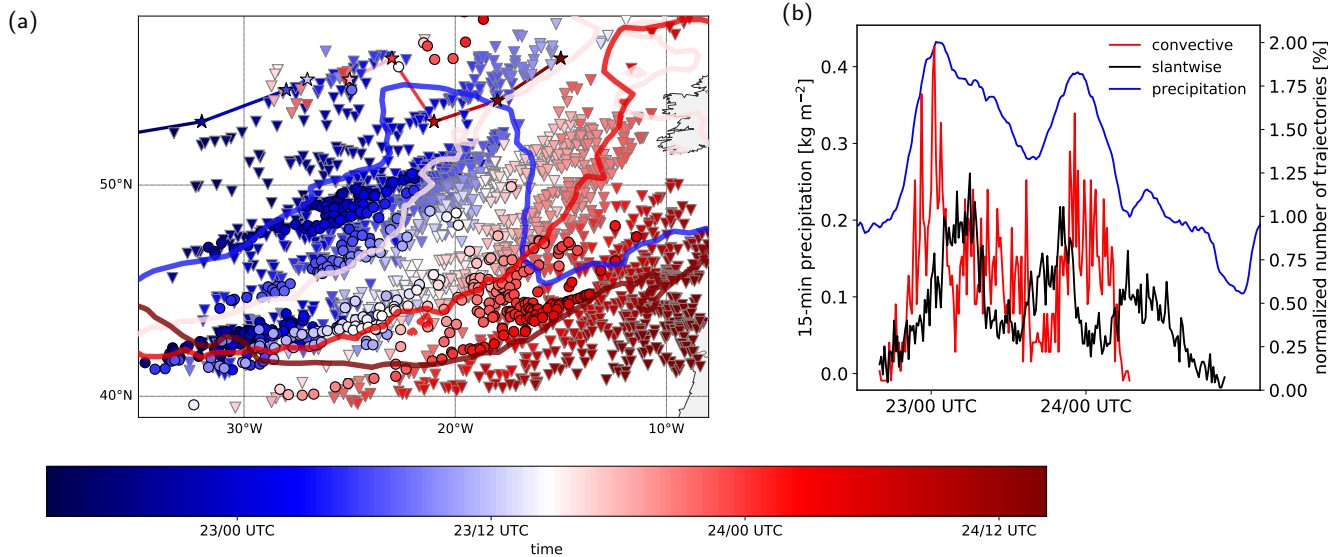

**Figure 3. (a)** Location of convective (black outlined circles) and slantwise (grey outlined triangles) WCB trajectories at the start of the fastest 400-hPa ascent phase. Colors indicate the according time of the WCB air parcel position (blue 23 Sep, red 24 Sep). Only every $4^{th}$ trajectory is shown. The evolution of the frontal structures is indicated by $\theta$=293 K at 850 hPa (lines are colored according to the selected times at 04 UTC 23 Sep, 16 UTC 23 Sep, 04 UTC 24 Sep and 16 UTC 24 Sep). The asterisks indicate the location of the surface cyclone every 6 h (colored according to time). **(b)** Temporal evolution of the number of selected convective (red) and slantwise (black) WCB trajectories at the start of the fastest 400-hPa ascent phase normalized by the absolute number of selected trajectories in each category and evolution of 15-minute accumulated precipitation averaged over the WCB domain shown in **(a)**. Note that for the domain-averaged precipitation only grid points with non-zero precipitation were considered.

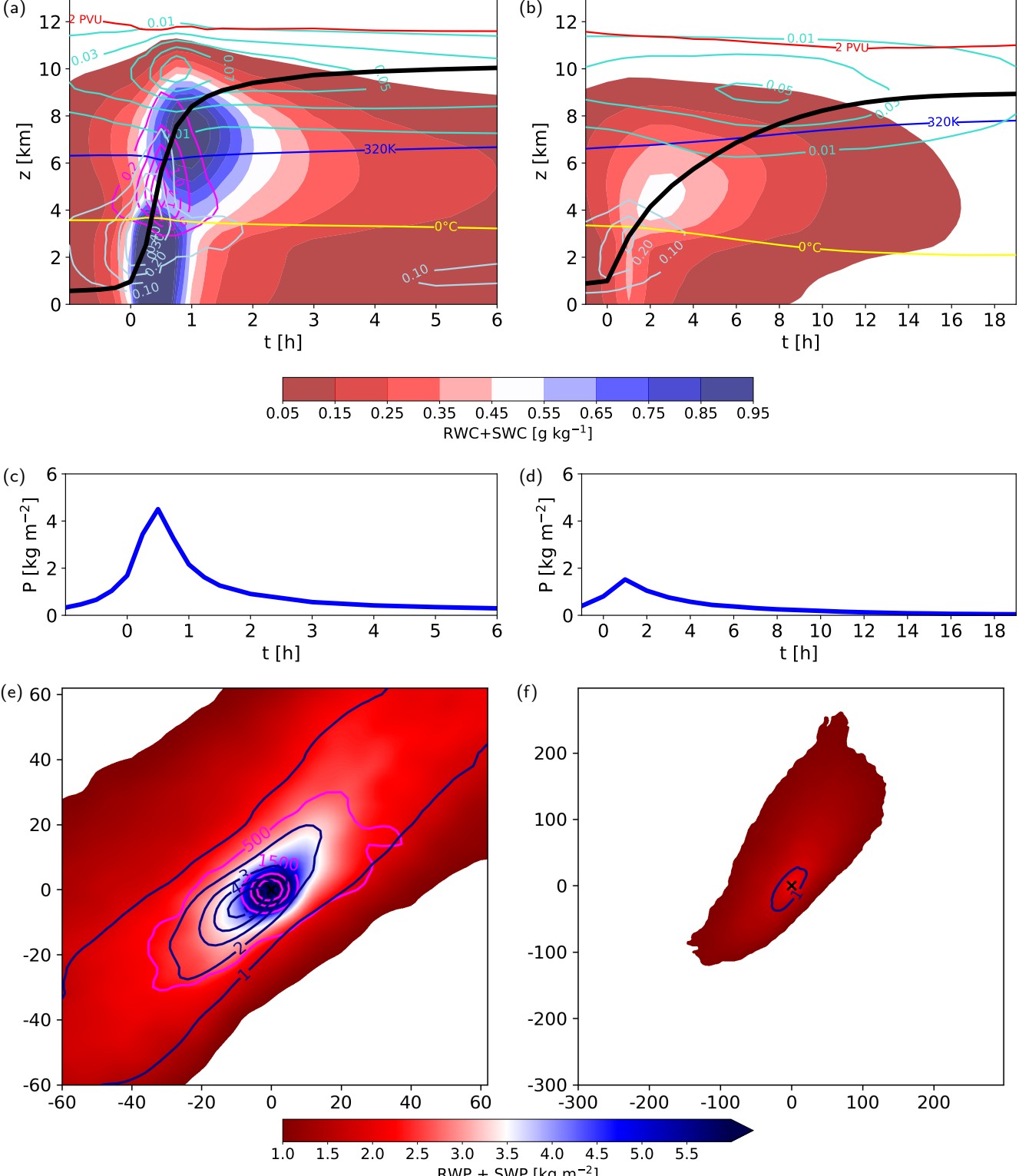

**Figure 4. (a,b)** Composites of vertical profiles following the motion of the trajectories (black line shows the mean ascent of all WCB trajectories) for **(a)** convective WCB trajectories and **(b)** slantwise WCB trajectories for hydrometeors [sum of rain and snow water content (RWC + SWC, colors, in $g\,kg^{-1}$), ice water content (IWC, turquoise contours, every $0.02\,g\,kg^{-1}$), liquid water content (LWC, light blue contours, every $0.1\,g\,kg^{-1}$), graupel water content (GWC, magenta contours, every $0.4\,g\,kg^{-1}$), 0°C-isotherm (yellow line), 320-K isentrope (blue line) and the 2-PVU tropopause (red line). **(c,d)** 15-minute accumulated surface precipitation along the ascent of the **(c)** convective and **(d)** slantwise WCB trajectories (blue line, in $kg\,m^{-2}$). **(e,f)** Horizontal cross-section composites of vertically integrated rain and snow water content (RWP + SWP, colors, in $kg\,m^{-2}$), vertically integrated graupel water content (magenta contours, every $500\,g\,m^{-2}$) and 15-minute accumulated surface precipitation (blue contours, every $1\,kg\,m^{-2}$) for **(a)** convective WCB trajectories 30 minutes and **(b)** slantwise WCB trajectories 1 h after the start of the fastest 400-hPa ascent (corresponding to the respective times of maximum surface precipitation). The axes' dimensions denote the distance from the WCB air parcel locations marked as '×' (in km). Note the different time axis of **(a,c)** and **(b,d)** and the different horizontal dimensions of **(e)** and **(f)**.

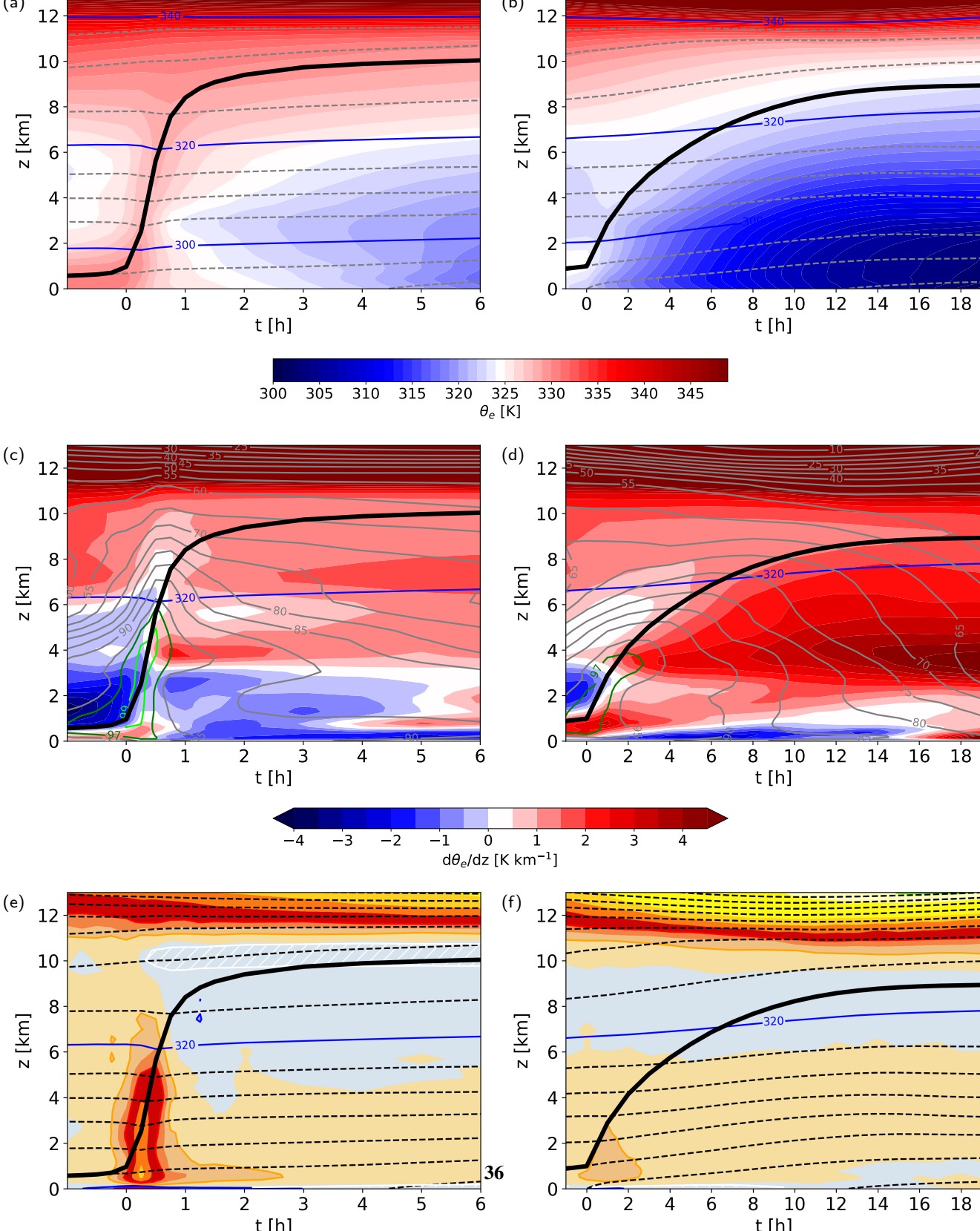

**Figure 5.** Composites of vertical profiles following the motion of the trajectories (black line shows the mean ascent of all WCB trajectories) for **(a,c,e)** convective WCB trajectories and **(b,d,f)** slantwise WCB trajectories for **(a,b)** equivalent potential temperature ($\theta_e$, colors, in K), potential temperature ($\theta$, grey dashed lines, every 5 K) and the 300, 320 and 340-K isentrope (blue lines); **(c,d)** moist stability ($d\theta_e/dz$, colors, in K km$^{-1}$), the 320-K isentrope (blue line), and relative humidity (RH, grey contours, in %; 97% and 99% RH contours are highlighted in green and lime); **(e,f)** PV (colors, in PVU), isentropes (dashed lines, every 5 K), the 320-K isentrope (blue line), and in **(e)** low static stability layers ($d\theta/dz \leq 2$ K km$^{-1}$, white contour and hatching). Note the different time axis in **(a,c,e)** and **(b,d,f)**.

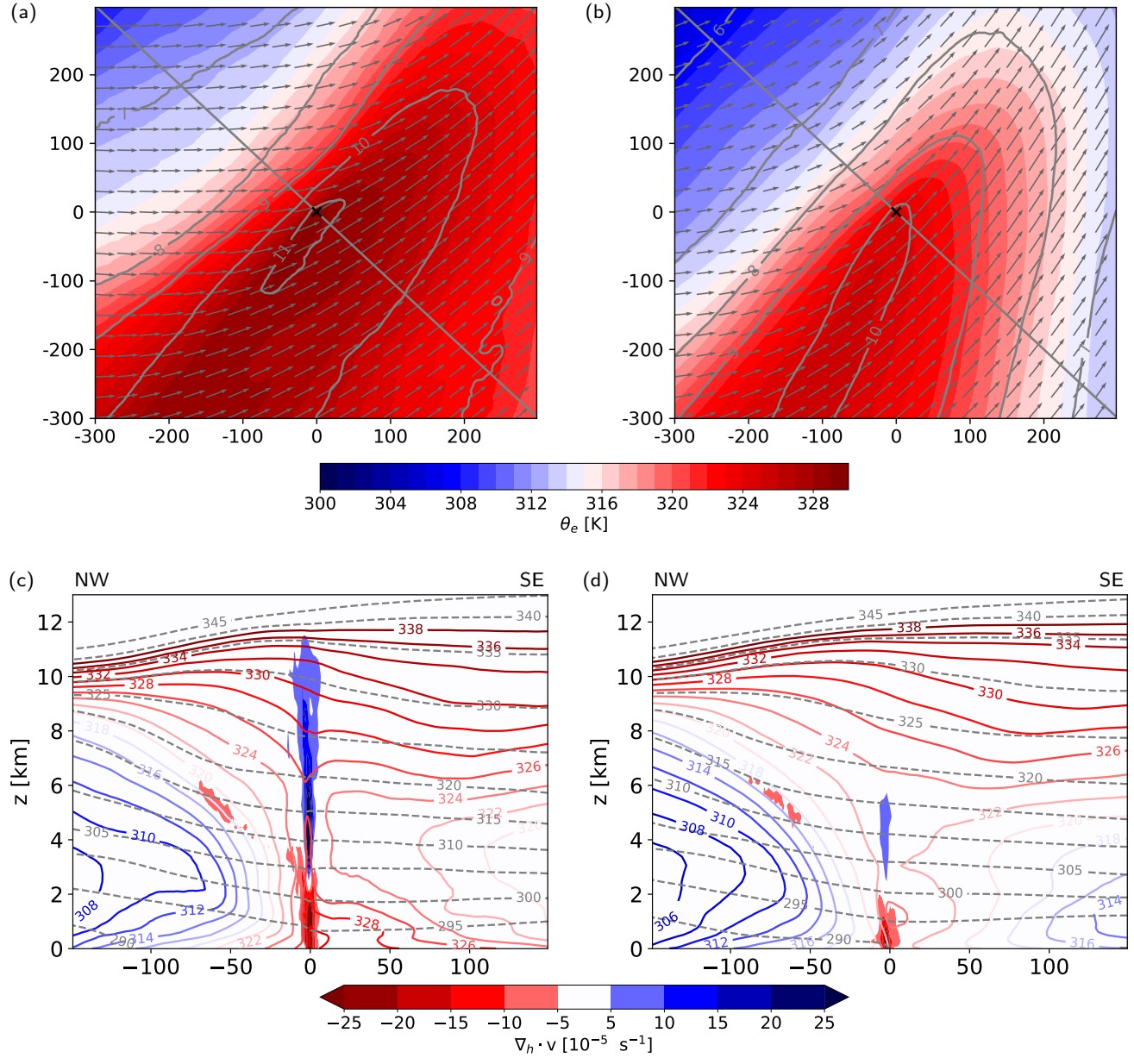

**Figure 6. (a,b)** Horizontal cross-section composites of $\theta_e$ at 900 hPa (colors, in K), specific humidity (grey contours, every $1\,\mathrm{g\,kg^{-1}}$) and wind at 900 hPa (arrows) for **(a)** convective WCB trajectories and **(b)** slantwise WCB trajectories at the start of the fastest 400 hPa-ascent. The axes' dimensions denote the distance from the WCB air parcel locations marked as '$\times$' (in km). **(c,d)** Vertical cross-section composite along the northwest-southeast orientated lines shown in **(a,b)** for horizontal wind divergence (colors, in $\mathrm{s^{-1}}$), equivalent potential temperature ($\theta_e$, red - blue lines, every 2 K) and potential temperature ($\theta$, grey dashed lines, every 5 K) for **(c)** convective and **(d)** slantwise WCB trajectories at the start of the fastest 400-hPa ascent. The $x$-axis denotes the zonal distance from the WCB air parcel locations (in km).

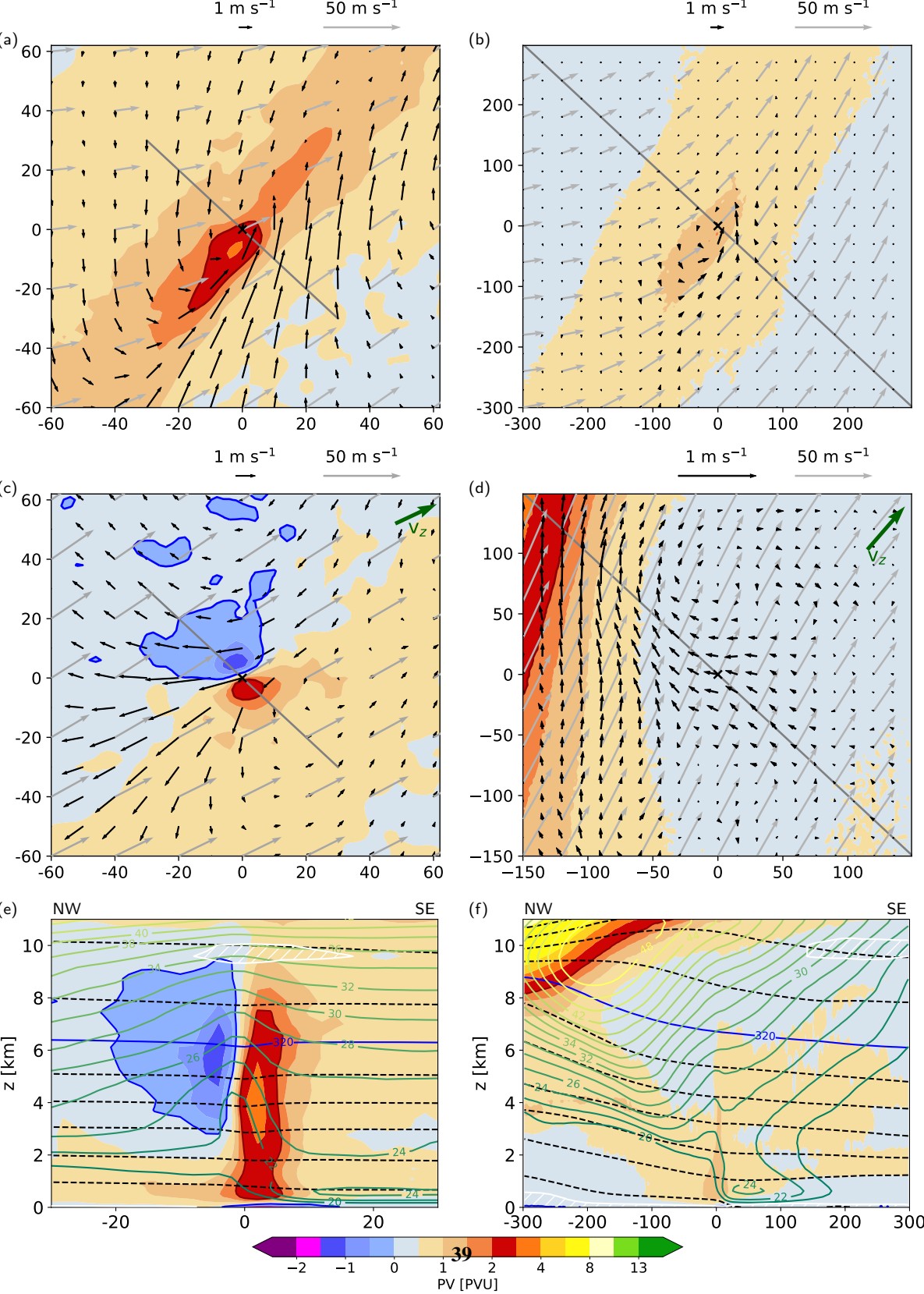

**Figure 7. (a-d)** Horizontal cross-section composites of PV (colors, in PVU), wind speed (grey arrows, in m s$^{-1}$) and 2-h circulation anomalies (black arrows, in m s$^{-1}$) at **(a,b)** 800 m and **(c,d)** 320 K for **(a,c)** convective WCB trajectories 30 minutes after the start of the fastest 400-hPa ascent, and for **(b,d)** slantwise WCB trajectories **(b)** 1 h and **(d)** 10 h after the start of the fastest 400-hPa ascent. The green arrow in **(c,d)** shows the direction of the vertical wind shear vector ($v_z$) between 4 and 10 km height at the location of WCB ascent. The axes' dimensions denote the distance from the WCB air parcel locations marked as '$\times$' (in km). **(e,f)** Vertical cross-section composites along the northwest-
southeast orientated lines shown in **(a,b)** of PV (colors, in PVU), wind speed (green contours, every 2 m s$^{-1}$), isentropes (dashed lines, every 5 K), the 320-K isentrope (blue line) and low static stability layers ($d\theta/dz \leq 2$ K km$^{-1}$, white contour and hatching) for **(e)** convective WCB trajectories 30 minutes and **(f)** slantwise WCB trajectories 1 h after the start of the fastest 400-hPa ascent. The $x$-axis denotes the zonal distance from the WCB air parcel locations (in km). Note the different spatial dimensions for the convective and slantwise WCB trajectories.

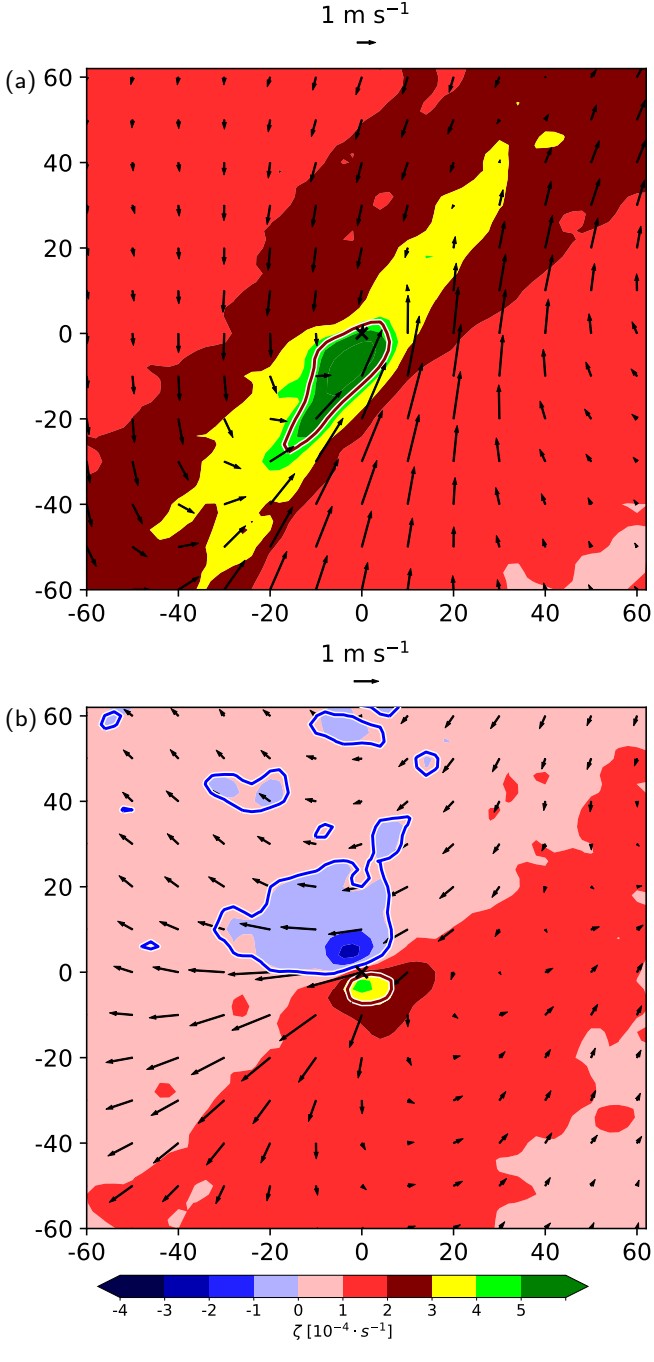

**Figure 8.** As Fig. 7a,c but for absolute vertical vorticity ($f + \zeta$, colors, in $\mathrm{s}^{-1}$) and 2-h circulation anomalies (black arrows, in $\mathrm{m\,s}^{-1}$) at **(a)** 800 m and **(b)** 320 K for convective WCB trajectories 30 minutes after the start of the fastest 400-hPa ascent. The 0 and 2-PVU contour lines are shown in blue and red, respectively.

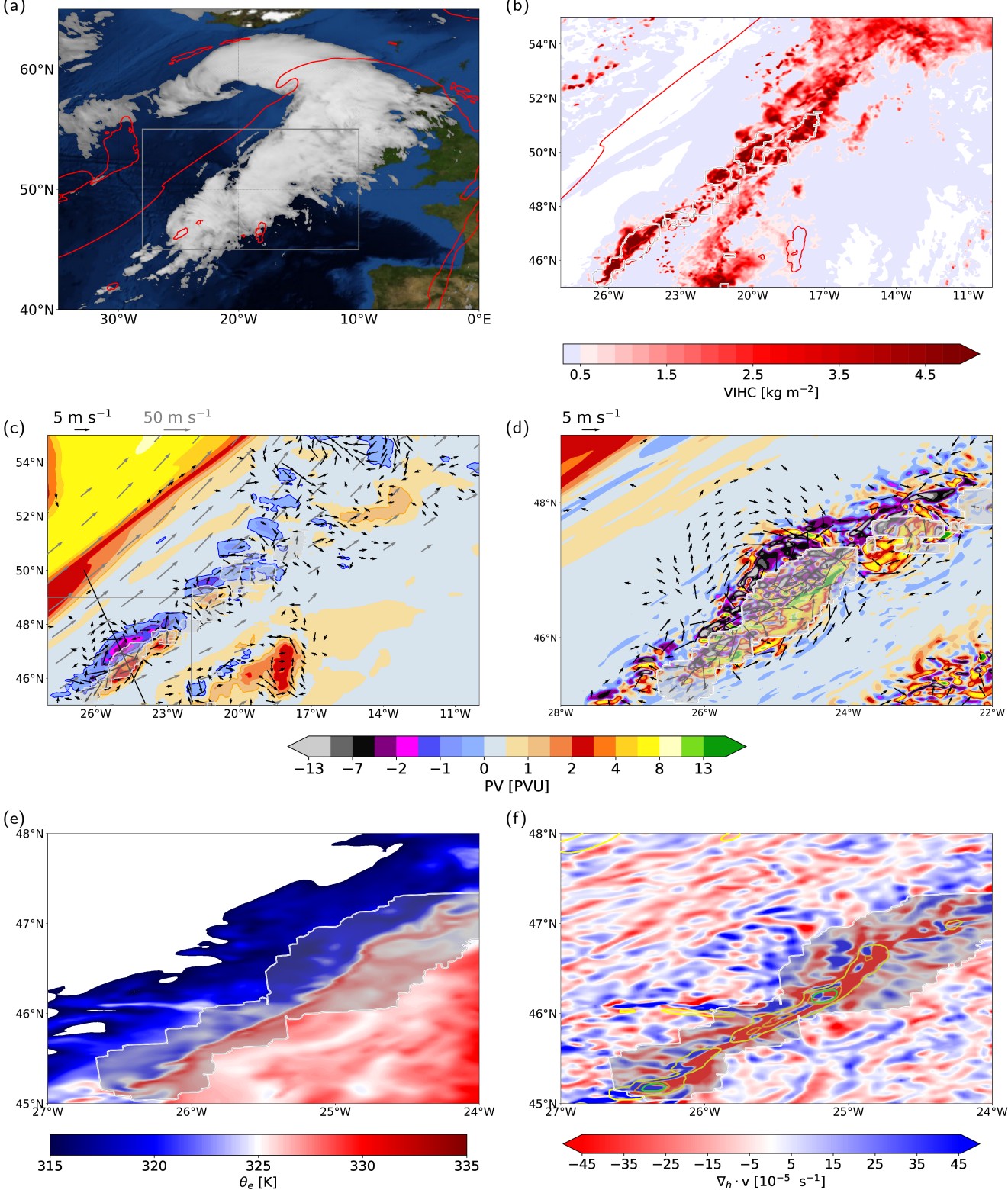

**Figure 9.** Example of embedded convection in the WCB at 09 UTC 23 Sep 2016. **(a)** Pseudo-IR satellite image of the large-scale cloud structure [data from 10.8 $\mu$m channel, EUMETSAT, Schmetz et al. (2002)] and 2-PVU contour at 320 K (red line); **(b)** vertically integrated hydrometeor content (VIHC, in kg m$^{-2}$, colors for VIHC>5 g m$^{-2}$) for the region outlined in **(a)** (grey box), envelope of rapid WCB ascent (white outline, WCB trajectory ascent >320 hPa in 2 h) and 2-PVU contour at 320 K (red line); **(c)** coarse-grained PV at 320 K (colors, in PVU; purple, blue, orange and red contour lines show -1, 0, 1 and 2 PVU; see text for details), 2-h circulation anomalies at 320 K (black arrows), wind speed at 320 K (grey arrows) and envelope of rapid WCB ascent (white outline); **(d)** enlargement of embedded convection shown in **(c)** (grey box) with PV at 320 K from the original 2-km model grid, 2-h circulation anomalies at 320 K (black arrows) and envelope of rapid WCB ascent (white outline); **(e)** enlargement of embedded convection shown in **(c)** with equivalent potential temperature at 900 hPa ($\theta_e$, colors, in K) and envelope for rapid WCB ascent (white outline); **(f)** as **(e)** but for low-level wind divergence at 295 K (colors, in s$^{-1}$), PV (coarse-grained to 12 km) at 295 K (yellow, lime and magenta contour lines at 2, 4, and 5 PVU) and envelope of rapid WCB ascent (white outline).

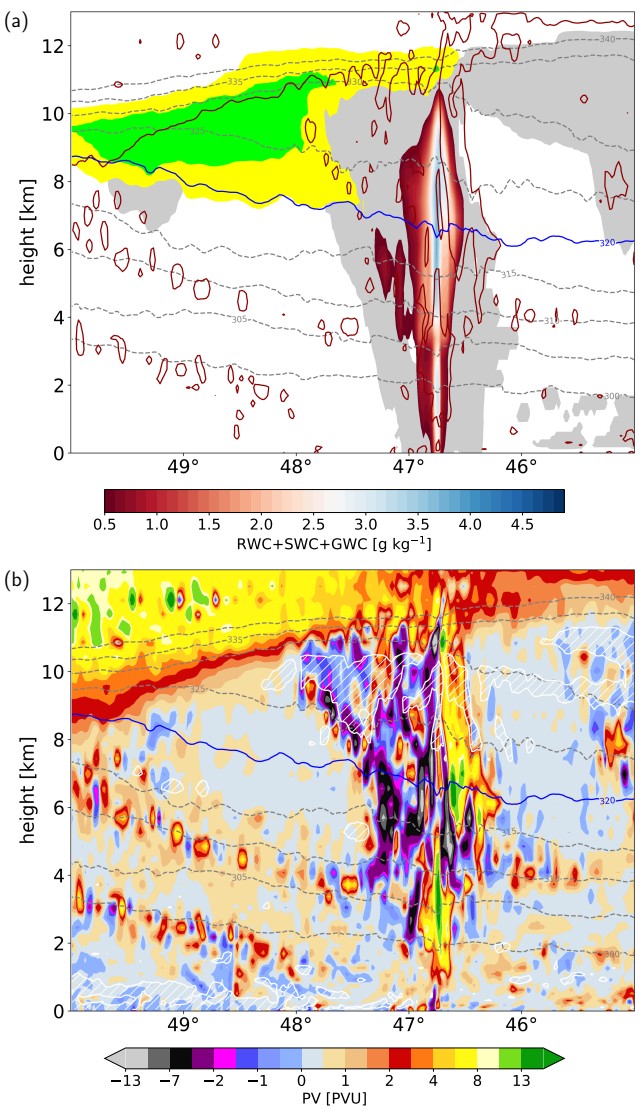

**Figure 10.** Vertical cross-section through the PV dipole shown in Fig. 9c (black line) at 09 UTC 23 Sep 2016 for **(a)** sum of rain, snow and graupel water content (RWC + SWC + GWC, colors, in g kg$^{-1}$), cloudy region (grey shading, hydrometeor content > 1 mg kg$^{-1}$), upper-level jet (yellow and green shading at 55 and 60 m s$^{-1}$) and 2-PVU contour (dark red line), and **(b)** PV (colors, in PVU) and low static stability layers (d$\theta$/dz ≤ 2 K km$^{-1}$, white contour and hatching). Isentropes (dashed contours, every 5 K) and the 320-K isentrope (blue line) are shown in both panels.

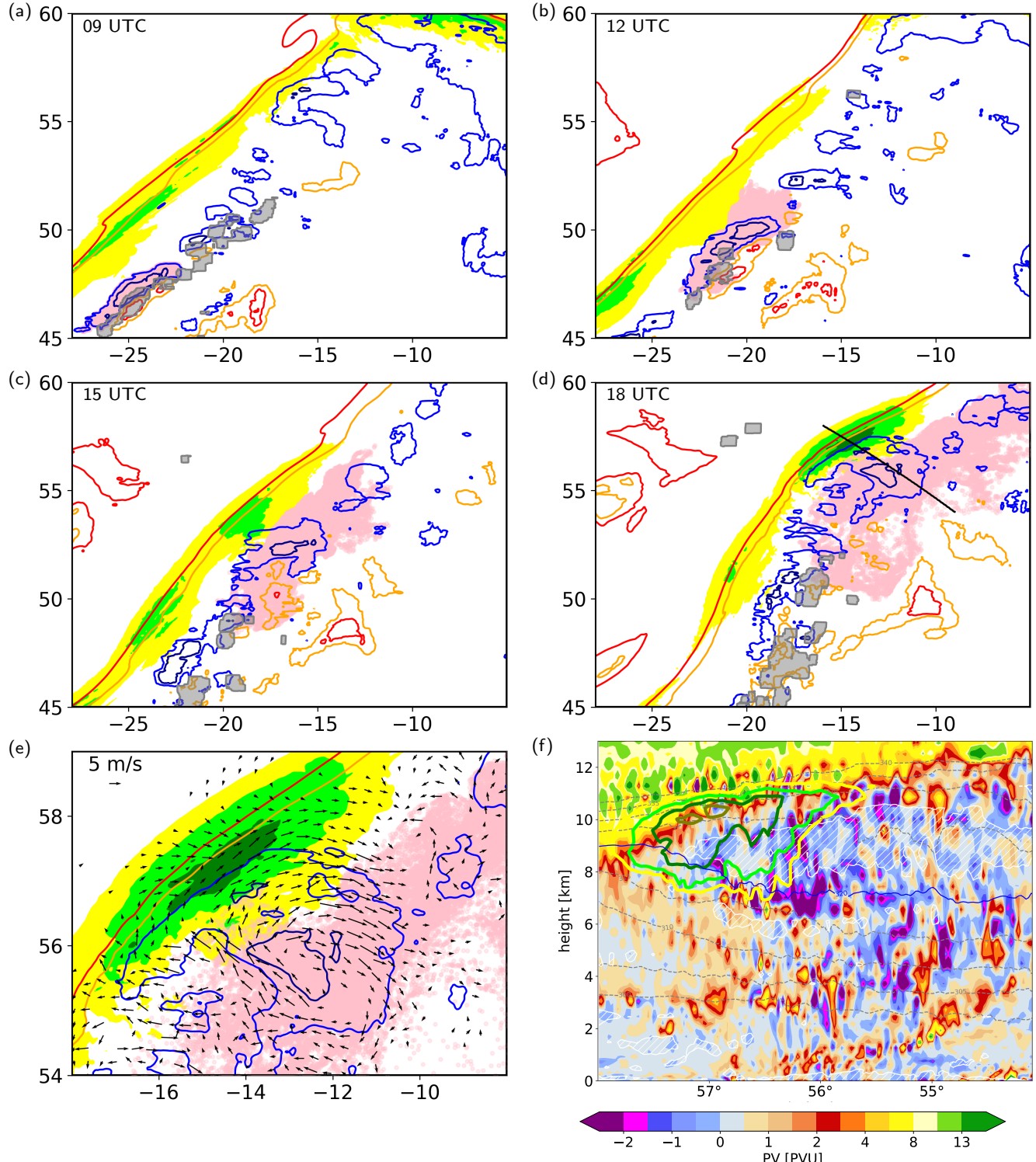

**Figure 11.** Spatially averaged upper-level PV at 320 K (dark blue, blue, orange and red contours at -1, 0, 1 and 2 PVU), upper-level jet at 320 K (yellow and green colors at 55, 60 and 65 m s$^{-1}$) and envelope of rapid WCB ascent (grey contour and shading) at **(a)** 09 UTC, **(b)** 12 UTC, **(c)** 15 UTC and **(d)** 18 UTC 23 Sep 2016. The pink shading shows the positions of forward trajectories initialized in a region of convectively produced negative PV between 315 and 325 K at 09 UTC; **(e)** is an enlargement of **(d)** and additionally shows the 2-h circulation anomaly (black arrows); **(f)** Vertical cross-section along the line shown in **(d)** for PV (colors, in PVU), potential temperature (grey dashed

lines, every 5 K), 320-K isentrope (blue line), and jet (yellow and green contours, at 55, 60, 65 and 70 m s$^{-1}$). Low static stability layers ($d\theta/dz \leq 2$ K km$^{-1}$) are indicated by the white contour and hatching.

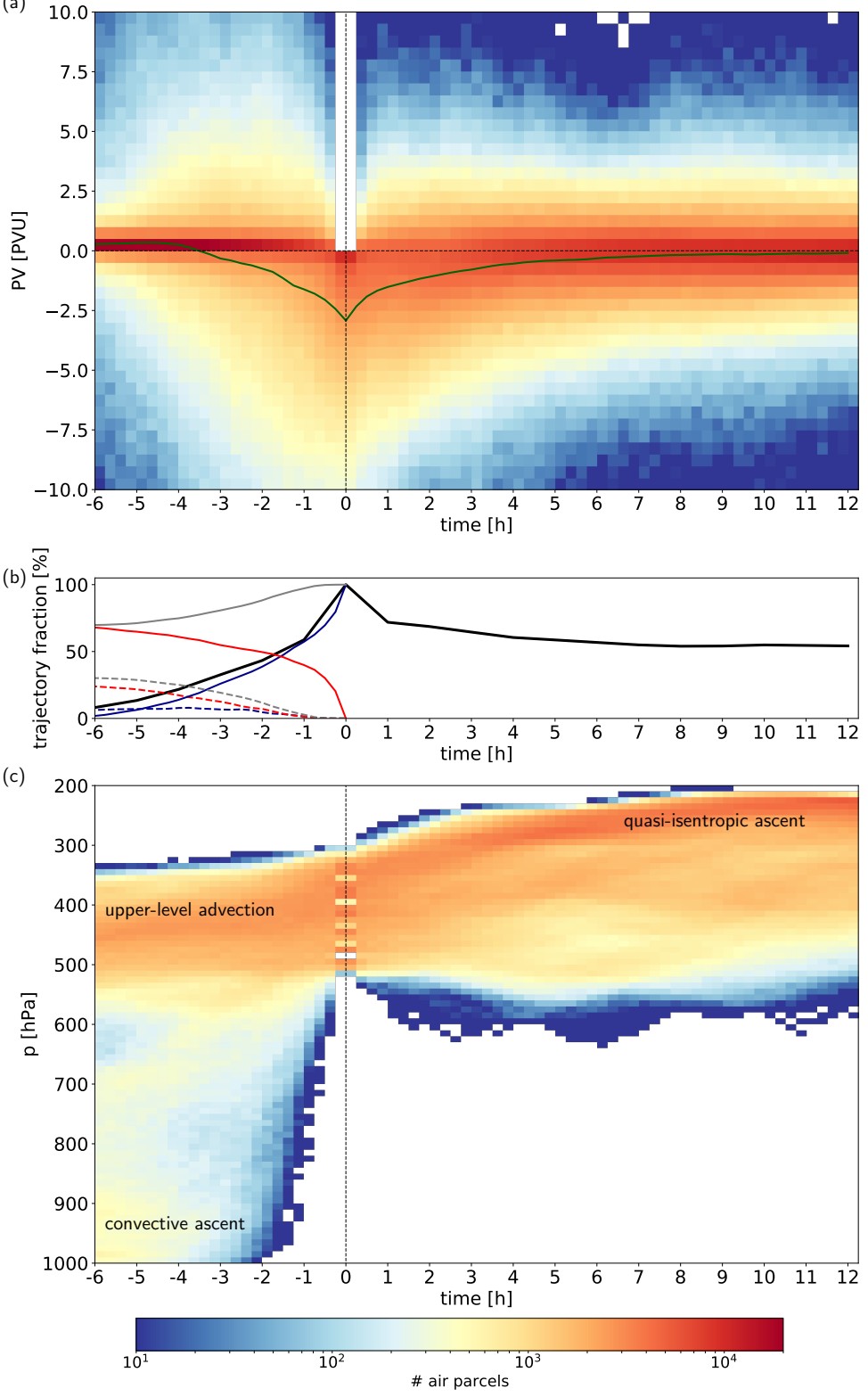

**Figure 12. (a)** 2D histogram of temporal evolution of PV (in PVU) for forward and backward trajectories initialized in a region of convectively produced negative PV between 315 and 325 K at 09 UTC as shown in Fig. 11a (pink shading). The time is given relative to the trajectory start at 09 UTC. The green line shows PV averaged over all trajectories (in PVU). **(b)** Fraction of trajectories with a negative PV value (black, in %), the fraction of trajectories located above 600 hPa (solid grey, in %) and below 600 hPa (dashed grey, in %). Additionally, the fraction of trajectories with negative PV above 600 hPa (blue solid, in %) and below 600 hPa (blue dashed, in %), as well as the fraction of trajectories with positive PV above 600 hPa (red solid, in %) and below 600 hPa (red dashed, in %) are shown. Note that after $t=0$ h the majority of trajectories is located above 600 hPa. **(c)** As **(a)** but for pressure (in hPa).

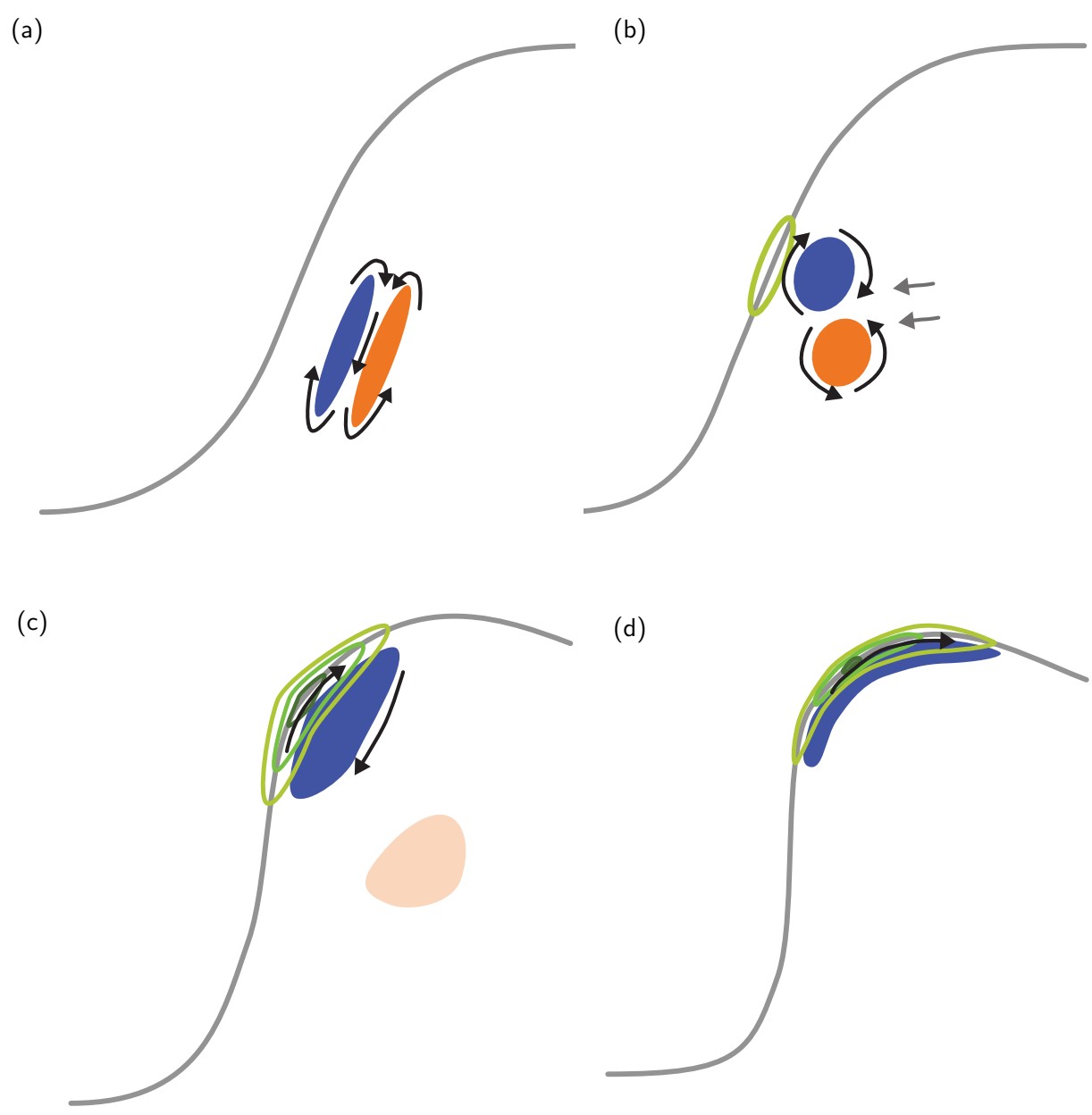

**Figure 13.** Schematic illustration of different stages of convectively generated PV (orange and blue colors for the positive and negative PV poles, respectively) and the associated jet structure (green contours); **(a)** PV dipole band; **(b)** meridional alignment; **(c)** distorted negative PV and weakend positive PV (light orange); **(d)** stretched negative PV band. See text for details.