# Peer review of "Potential vorticity structure of embedded convection in a warm conveyor belt and its relevance for the large-scale dynamics"

_Weather and Climate Dynamics, 2019_

## Referee Comment (RC1) · Florian Pantillon (Referee) · 17 Oct 2019

Review of "Potential vorticity structure of embedded convection in a warm conveyor belt and its relevance for the large-scale dynamics" by Annika Oertel, Maxi Boettcher, Hanna Joos, Michael Sprenger, and Heini Wernli.

The paper presents a detailed analysis of a case study of warm conveyor belt associated with an extratropical cyclone observed during the NAWDEX field campaign and described in a previous paper. Thanks to Lagrangian trajectories computed online in a convection-permitting simulation of the case study, convective ascents are distinguished from slantwise ascents within the warm conveyor belt. A composite analysis

highlights the warmer and moister origin, the production of graupel and more intense precipitation for convective ascents. A horizontal dipole of potential vorticity in the outflow region is explained theoretically by considering the evolution of horizontal and not only vertical vorticity components. An example of frontal convection is then described to illustrate the formation of a dipole band of potential vorticity and its evolution suggests a contribution to upper-level gradients and jet acceleration.

The paper addresses the important question of how small-scale convective processes to large-scale dynamical processes, which is often neglected due to the inability of most NWP model runs to encompass both scales. The results are based on relevant methods, sound interpretation and high-quality figures. However, the paper is unbalanced between, one the one hand, a composite analysis of all convective ascents to investigate their dynamics and, on the other hand, an illustration of a specific convective ascent to investigate its impact, which implications tend to be over-interpreted. Moreover, the reasoning is often difficult to follow due to breaks in the logic flow, jumps from one figure to the other, and statements that lack precision. Several general and numerous specific comments are listed below to help improving the structure and clarity of the paper.

GENERAL COMMENTS

I. Unlike the systematic analysis of convective ascents based on composites, the impact on the large scale is investigated based on a single convective PV dipole band, is rather qualitative and uses offline instead of online trajectories. Either extend toward a more quantitative framework, e.g. also based on composites, or at least carefully discuss the results and implications throughout the paper (including in title and abstract) considering these limitations.

II. The "big picture" is diluted in the introduction: the contrast between small and large scale is clear but what is referred to as mesoscale? There is a confusion between isolated cells and organized convection, and embedded convection needs a proper

definition (from the example in Fig. 8 it appears to develop along the cold front only). Citations need to be revised and the organization should be improved (see next comment).

III. Theoretical considerations on PV production need reorganization: they appear too early in the introduction and are largely repeated in Sections 3.4.3 and 3.4.4. Consider moving the detailed PV discussion to a short theoretical section, which could later be referred to. Some elements may be moved from the discussion section and grouped with either the introduction or the theoretical considerations.

IV. Most of the paper is based on composites of Lagrangian trajectories, which is a very interesting approach but may look abstract to the reader; sometimes a figure in geographical coordinates would be helpful. Placed earlier, Section 4 would well introduce the case study and illustrate the different concepts that are developed in sections 3 and 5, thus remove the need to constantly refer to Oertel et al. 2019 and to Fig. 8(d) early in the paper.

V. The text contains several repetitions and frequently jumps back and forth between figures. In addition, it often refers to Oertel et al. 2019 and other papers to explain results, which creates a confusion between what is expected from previous work and what is actually found here. Please streamline and clarify to improve the flow.

SPECIFIC COMMENTS

Abstract

l. 4-5 not sure this sentence is needed

l. 8-10 the sentence somehow suggests that graupel is part of surface precipitation

l. 11 what does "they" refer to?

l. 17 perhaps insist on negative PV values? (not just anomaly)

l. 17-19 this is speculative, as only one example is presented

l. 21-23 this is also speculative, for the same reason

l. 23-25 this basically repeats what is written above; what are broader implications of the study?

Introduction

l. 27 "their"?

l. 29 The two references are not clearly related to moist diabatic processes in extratropical cyclones.

l. 35 "potentially affect": better "can affect"?

l. 37 The transition from general WCB dynamics to precise PV theory is abrupt.

l. 42 Please define terms in brackets (or omit).

l. 46 Please define f, zeta and theta_dot.

l. 47-49 This sounds very similar to l. 34-36.

l. 51 The first two references do not mention mesoscale convective systems.

l. 52 What are "convective storms"? (vs. MCS above)

l. 58-59 Is this not what has just been stated in l. 53-57?

l. 61 The horizontal vorticity vector w_h must be defined when it is introduced first.

l. 60-63 This process is not obvious and requires more details. I appreciate it is supported by a schematic but I do not exactly understand what is what in Fig. 1. Can you make the schematic more reader-friendly, e.g. by illustrating why the vectors are oriented as they are, using more explicit colors and referring to them in the text?

l. 71-75 There is a confusion between negative PV, the different types of instabilities and their consequences. Schultz and Schumacher (1999) rather discuss conditional symmetric instability, which is another type of (slantwise) instability.

l. 79 "observed" is not the appropriate word for model data. Better "found"?

l. 81-93 This general paragraph would rather better appear before the previous, more specific one.

l. 87 The cited studies use very different types of "remote-sensing data". More specifically?

l. 89-93 There is a confusion between what impact and which study relates to WCB or convective systems. Furthermore, the link with forecast errors would better fit with the modification of the large-scale flow in the first paragraph.

l. 94-114 Rather than pointing what has not been done yet and focusing on very precise questions, this part would better motivate the study if it would emphasize what are open questions (e.g., contribution of embedded convection to WCB dynamics), why they are important (e.g., reconcile small- and large-scale views) and how they are addressed here (e.g., convection-permitting simulations of a case study).

l. 94 MCSs are not just "individual convective updrafts".

l. 116 which case study?

Data and approach

Past and present tenses are often mixed in this Section, please stick to one or the other.

l. 123 is it the same setup or the same simulation?

l. 125 "grid spacing" rather than "resolution".

l. 142-158 It is difficult to get a general picture of the cyclone evolution without reading Oertel et al. (2019). A graphical summary would be helpful, e.g. by adding the cyclone track (and the upper-level trough) on Fig. 2(a).

l. 142 explicit NAWDEX

l. 150 more than 400 or 600 hPa?

l. 156-157 this last sentence is unnecessary

l. 163-166 Why combine two criteria? (400 hPa in 1 h and 600 hPa in 3 h) Is a fast ascent (top 10%) sufficient to be considered "convective"?

l. 170-172 The description is confusing: temporal evolution of what? is it really the position relative to the cold front? Where is the upper-level trough?

l. 174 behind rather than "ahead of"?

l. 176 again, showing the cyclone position would be helpful for the general picture.

Characteristics of convective and slantwise WCB ascent

l. 183-184 what about the vertical coordinate?

l. 188-189 not only the impact but also the environment of trajectories

l. 190 is this shown somewhere?

l. 193-194 does it mean that circles in Fig. 2(a) are also at 00 UTC 23 and 24 Sep mainly?

l. 201-204 "warmer and moister region": is it really warmer? Fig. 3(a,b) does not explicitly show the contribution of theta to theta_e and Fig. 4(a,b) suggests that the difference in absolute temperature T is due to a different height. Also, some indication about how much the two composites overlap is needed, either by showing statistical significance on cross-sections at least by giving the standard deviation around mean values.

l. 205-211 apart from highlighting low-level convergence, it seems that these lines do not add any new information to "the region is warmer and moister"; clarify or streamline.

l. 214-215, 222-223 what about the difference in height? Is it significant?

l. 218-219 can you be more precise about where to find this information in Oertel et al. 2019?

l. 221 "the observed rapid convective updrafts" and associated references: it sounds like a conceptual description of potential instability but not necessarily of what happens here

l. 222 mention somewhere the different time scale to emphasize the different ascent rate

l. 226-228 can you be more precise by giving a value of attained height (average+/-std)

l. 230 Please design new panels for the time evolution of surface precipitation in Fig. 5; panels (c,d) are already very busy and mixing vertical profiles with a scalar value is extremely confusing. It should also be mentioned somewhere that (a,b) are instantaneous values taken at the respective time of max surface precipitation.

l. 236 "comparatively thick cirrus cloud" compared to what?

l. 237 turquoise contours?

l. 237-241 it is unclear how the cirrus cloud related to convection, as its core is located well above the composite trajectory; is it due to a fraction of faster-ascending trajectories?

l. 243-244 "horizontally more homogeneous": can this really be seen in time-height plots?

l. 246-247 this is interesting indeed, but may it be due to the compositing process, or are there actual profiles where ice water extends above the tropopause level?

l. 251-253 this largely repeats what is written above and is thus unnecessary

l. 255-256 number of trajectories starting their ascent at that time?

l. 256 is the precipitation area roughly constant, i.e., are variations in Fig. 2(b) due to

variations in intensity or in concentration?

l. 257 "Nevertheless": furthermore?

l. 258-259 what is the citation here needed for? Clarify or omit

l. 278-282 where is this effect seen in Fig. 5(c,d)?

l. 286 "observed PV distribution": more specifically? Avoid "observed" if from model

l. 294-295 "in particular" seems to contradict "despite" above

l. 301-302 why that time? (maximum precipitation rate?)

l. 303-306 mention the different scales in (a,b) or add box of (a) in (b)?

l. 308-309 "as a consequence" appears to repeat "due to"

l. 317 Fig. 5(c) does not explicitly show diabatic heating

l. 322-324 this statement appears speculative

l. 323 specify what to look at in Oertel et al., 2019, Fig. A1

l. 330-332 I do not clearly see the vertical PV dipole expected in this case according to the previous sentence

l. 345-346 how is this shown in Fig. 7(a)?

l. 346-350 is it an interpretation or is it really shown somewhere?

l. 351-358 similar to above, is this shown somewhere for the composite or does it refer to the theoretical schematic only?

l. 359-366 This largely confirms what is explained in the introduction

l. 368 again, Schultz and Schumacher (1999) mainly discuss conditional symmetric instability

l. 382-384 this partly repeats l. 373-375

l. 395-398 does it occur here? Is it shown anywhere?

l. 402 "is accelerated by" contradicts "hardly exceeds" above

l. 406 "PV dipoles" plural or singular?

l. 407-409 for comparison, what is the value of the vertical shear?

An illustrative example of WCB-embedded convection

The purpose of this section is unclear at that point, as it mostly repeats ideas developed in the previous section; such an "illustrative example" would better fit early in the paper to motivate the systematic analysis based on composites.

l. 420 at 09 UTC 23 Sep 2016

l. 422-423 the previous section insists on the presence of graupel to distinguish convective from slatwise ascent: display graupel here only? And does it occur along the cold front?

l. 423-424 "rapidly ascending WCB trajectories": convective WCB trajectories?

l. 427-429 this last sentence mostly repeats what has just been stated

l. 432 is PV on the original grid or aggregated in the cross-section?

l. 434 PV below -2 PVU cannot be seen with the colour bar; horizontal PV gradients?

l. 435-436 is the heating maximum shown somewhere?

l. 436 "lens" without e

l. 437-438 please motivate the statement and clarify "mesoscale PV dipole"

l. 441 "rapid WCB ascent": convective WCB trajectories?

l. 445-446 "which are generated and further enhanced by convective ascent" sounds

speculative

l. 459-450 is the thermal wind vector shown somewhere?

l. 458-464 This belongs to the introduction

l. 470 northwest

PV anomalies on a larger scale and relevance for large-scale dynamics

l. 478 is this supported by section 3 (for this case) or by Shutts 2017 (in general)?

l. 479-480 remove "these"

l. 480-483 this sounds as three times the same statement, clarify or streamline; "effective resolution" has a specific meaning for numerical modeling, better avoid

l. 485 is this the case for all larger-scale PV anomalies, or for the example of section 4 only?

l. 486 is the cold front shown somewhere?

l. 490 this seems to describe a specific feature rather than "PV dipole bands"

l. 491 southeastward

l. 492 repetition of earlier statements

l. 493-497 what is seen where? (which contour, colour, panel)

l. 502-509 more arguments are needed to support that the convectively-produced PV dipole in Fig. 10(a) evolves into the anticyclonic anomaly in Fig. 10(e): the trajectories spread over a much larger area than this specific feature at 18 UTC, and other PV structures exist during the evolution

l. 514 why use offline trajectories, while online trajectories better follow convective ascent?

l. 517-518 this largely repeats the previous sentence

l. 525-534 the paragraph contradicts the last sentence in l. 523-524 and is confusing altogether; please clarify

l. 538 how exactly do parcels "gain negative PV"?

l. 544 indeed, a comparison with online trajectories is needed to support this result; but again, why use offline trajectories here?

l. 565 what should be compared between these figures? (which contours)

l. 569-570 this is not sufficiently supported; develop or omit

Discussion and open question

l. 575 not only one case study but one single PV dipole within a cyclone; a first step would be to look at other structures within this cyclone

l. 568 avoid "observations" if model-based

l. 586-594 these various impacts of embedded convection appear speculative; please clearly distinguish between what is due to convectively-generated PV anomalies and to the WCB outflow in general, and be precise about what the cited studies have shown

l. 607 heating is also parameterized, even at convection-permitting resolution, through the microphysical scheme

l. 611-612 "a horizontal resolution of at least 10 km would be required to resolve the convective updrafts": rather a grid spacing of a few km mostly, as in your simulation

Figure captions: "shading" better than "colours"

Providing titles to subfigures would be helpful, as most display rather complex content

Fig. 2(a) is too busy: consider showing less trajectories (every second, fifth, tenth, . . .) and one representative, thicker theta contour per lead time. It took me a while

to understand what is depicted and I still do not fully see the position of trajectories relative to the cold front.

Fig. 4 are these really "Vertical cross-section composites"? l. 186-187 rather refers to "composites of vertical profiles along the trajectories, i.e. time-height sections along the flow"; (a,b) 300, 320 and 340-K isentropes; (c,d) "(moist-adiabatic) lapse rate" rather than "potential instability"; d_theta/dz or d_theta_e/dz?

Fig. 5 "As Fig. 3a,b": not really, better explain again; what do RWP, SWP, RWC, SWC, . . . stand for? Check units; plot (a) box on (b) for comparison?

Fig. 8 this figure does not meet the otherwise high quality standard of the paper: tick-marks are too small and need °N/°E to indicate geographical coordinates (in contrast to km in composites); white contours are hardly seen on panels (b-d); vectors and vector legends are too small on (c-d); the colour bar is not adapted to the noisy field in (d); colour bars are completely saturated for negative values in (d-f); finally, (a) is not standard infrared imagery, what does it show exactly?

---

## Referee Comment (RC2) · Jeffrey Chagnon (Referee) · 29 Oct 2019

Summary:

This paper presents an analysis of the potential vorticity (PV) structures that accompany ascending air particles in a warm conveyor belt (WCB) simulated in the COSMO model. The analysis features a partitioning of WCB trajectories into "slantwise" and "convective" groups distinguished by their rates of ascent. A compositing technique is applied to identify the dynamical and thermodynamical structures associated with each of these groups. The convective air particles are associated with large PV dipole anomalies in the upper troposphere that are oriented horizontally with the negative

pole located to the left of the thermal wind. In contrast, the slantwise air particles are associated with much weaker PV anomalies. It is also shown that the convectively-generated negative PV anomalies comprise a coherent elongated region of low PV that is transported poleward towards the periphery of the jet.

This paper will make a valuable contribution to the literature. Of particular importance is the identification of the convective PV structure as well as its potential for influencing the synoptic scale. While previous studies have described convectively-generated horizontal PV dipoles, this paper is, to my knowledge, the first to demonstrate their importance in the context of extratropical cyclone dynamics. The results also have implications for numerical weather prediction (NWP); convection is difficult to simulate accurately, and if convection embedded in the WCB can have upscale influences on the synoptic evolution, then it is possible that convective-scale errors could degrade medium-range forecasts.

Overall, the paper is very well written. The analysis was conducted with meticulous attention to detail. I can find no major errors, but I have a few comments on the interpretation of results which I express below.

General Comments:

1. The headlining results in this paper concern the different behavior of convective versus slantwise trajectories. It is presumed that the convective nature of one group of particles is responsible for the deeper, larger-amplitude, coherent PV structures that accompany those particles. I would like to offer an alternative perspective for the authors to consider. In addition to being distinguished by their rate of ascent, the two groups of trajectories are also located in different regions of the WCB at their time of maximum ascent; specifically, the convective particles are located equatorward of the slantwise particles. The environments in which these two groups of particles ascend may therefore be different. Could the differences in environmental shear be primarily responsible for the different PV dipole structures? Is it possible that the shear vector

is oriented parallel to the front on the equatorward end of the front, whereas the shear vector is directed across the front on the poleward end? If so, then the PV dipoles should straddle the front on the equatorward end, whereas on the poleward end of the front they should be oriented along the front. According to this view, when compositing is performed, the dipoles on the equatorward side should retain a large amplitude PV dipole structures since there is less variance in the cross-frontal structure. On the other hand, the dipoles on the poleward side are subject to interference from neighboring dipoles along the front, resulting in a weaker PV structure in the composites. I suspect that the convective nature of the particles and the environmental shear are both important in determining the amplitude and structure of the PV anomalies.

2. While I agree with the authors contention that the horizontally-oriented PV dipoles are most likely due to heating in the presence of background shear (e.g. Figure 1), an alternative explanation for the structures (e.g., in Fig. 6e) is that they are associated with a vertically-oriented dipole in PV tendency that is subject to non-linear advection in the cross-frontal plane that results in the horizontally-oriented dipoles in total PV that we see. Have the authors examined either the PV tendencies or the cross-frontal advection to eliminate this possibility?

Specific comments:

1. In addition to calculating the composite mean maps, have the authors analyzed the variance? Variance maps could establish whether the mean maps are robust. For example, where the composite mean amplitude is large but the variance is low, the mean fields could be considered robust.

2. Figure 2 gives the impression that there is a large separation in time and space between the slantwise and convective particles, but the figure only shows the times and locations of maximum ascent. At any fixed time, would these groupings of particles occupy distinct regions of the WCB, or are they distributed more uniformly?

3. Line 169. Does the 2000 to 7000 split in the number of convective versus slantwise

particles imply that 3.5 times more mass ascends slantwise?

4. In Figure 5c, it is very difficult to distinguish the thick black lines from the thick blue lines.

Technical corrections:

1. Line 277. Should SWC and RWC be swapped?

2. Line 426. "is" -> "are"? (or make "hydrometeors" singular?)
* * *

---

## Author Comment (AC1) · 15 Jan 2020

**Manuscript wcd-2019-3**

**'Potential vorticity structure of embedded convection in a warm conveyor belt and its relevance for the large-scale dynamics'**

Oertel, A., Boettcher, M., Joos, H., Sprenger, M., and Wernli, H.

**Final author comments**

We would like to thank both reviewers, Florian Pantillon and Jeffrey Chagnon, very much for their positive, detailed and constructive feedback that helped to further improve the quality of this manuscript. We tried to address all the comments by the reviewers. The major changes in the new version of the manuscript are the following:

- 1. We restructured the introduction and parts of the results section and streamlined the text to improve its quality and to avoid jumping between figures too often.
- 2. We added a new overview figure to introduce the WCB case study (in geographical coordinates) and its embedded convection to avoid too many references to Oertel et al. (2019) and to improve the flow of the manuscript. Additionally, we will include two animations of PV at 320 K and the detailed WCB trajectory ascent in the online supplementary material.
- 3. The quality of the figures (essentially Figs. 6 and 8) was improved.
- 4. We added additional explanations in section 5 about the use of offline trajectories and our interpretation of the influence of the negative PV band on the larger-scale flow. A general comment by Florian Pantillon concerned the analysis of only one example in section 5. Our conclusions are based on the analysis of several of such PV dipole bands (which are, however, not shown in the manuscript). We provided two more examples in this document (Figs. 3 and 4 in this document) and mention it in the revised manuscript. We now also discuss the limitations of our analysis more clearly.
- 5. Finally, we slightly restructured the discussion and added a new figure in the discussion section that schematically outlines the concept of the interaction of the negative PV band with the upper-level trough.

Below are the detailed replies to the individual comments.

**1 Response to Jeffrey Chagnon**

**Comments to the author**

**General comments**

1. General comment 1

The headlining results in this paper concern the different behavior of convective versus slantwise trajectories. It is presumed that the convective nature of one group of particles is responsible for the deeper, larger-amplitude, coherent PV structures that accompany those particles. I would like to offer an alternative perspective for the authors to consider. In addition to being distinguished by their rate of ascent, the two groups of trajectories are also located in different regions of the WCB at their time of maximum ascent; specifically, the convective particles are located equatorward of the slantwise particles. The environments in which these two groups of particles ascend may therefore be different. Could the differences in environmental shear be primarily responsible for the different PV dipole structures? Is it possible that the shear vector is oriented parallel to the front on the equatorward end of the front, whereas the shear vector is directed across the front on the poleward end? If so, then the PV dipoles should straddle the front on the equatorward end, whereas on the poleward end of the front they should be oriented along the front. According to this view, when compositing is performed, the dipoles on the equatorward side should retain a large amplitude PV dipole structures since there is less variance in the cross-frontal structure. On the other hand, the dipoles on the poleward side are subject to interference from neighboring dipoles along the front, resulting in a weaker PV structure in the composites. I suspect that the convective nature of the particles and the environmental shear are both important in determining the amplitude and structure of the PV anomalies.

**Reply** Thanks for providing this alternative perspective on the PV signature of the slantwise WCB trajectories. While we agree that environmental shear is crucial to determine the amplitude, structure and orientation of the PV anomalies, we think that for the case of "Vladiana" the wind shear is not the relevant mechanism causing the observed differences between both categories. The analysis of the vertical wind shear profiles (Fig. 1 in this document) suggests that both the magnitude of the vertical wind shear and its direction are not substantially different for the ascent regions of the convective and the slantwise WCB trajectories. We rather think that in this case, the horizontal heating gradients ( $\nabla_h \dot{\theta}$ ) are weaker for the slantwise WCB trajectories because the rapid and localized convective ascent leads to more localized and stronger horizontal diabatic heating maxima (in agreement with the very localized increased hydrometeor production). The larger-scale slantwise WCB ascent, in contrast, produces weaker horizontal heating gradients. We added the direction of the vertical wind shear vector in the revised manuscript (green arrows in Fig. 7c,d).

2. General comment 2

While I agree with the authors contention that the horizontally-oriented PV dipoles are most likely due to heating in the presence of background shear (e.g. Figure

Figure 1: Vertical wind speed profile (mean and standard deviation) and direction of standardized vertical wind shear vector at selected altitudes for (a) the convective WCB trajectories 30 minutes and (b) the slantwise WCB trajectories 6 hours after the start of their fastest ascent phase.

1), an alternative explanation for the structures (e.g., in Fig. 6e) is that they are associated with a vertically-oriented dipole in PV tendency that is subject to non-linear advection in the cross-frontal plane that results in the horizontally-oriented dipoles in total PV that we see. Have the authors examined either the PV tendencies or the cross-frontal advection to eliminate this possibility?

**Reply** Unfortunately, we cannot output the PV tendencies in our simulation and we did not specifically consider the cross-frontal circulation. Nevertheless, we believe that the described mechanism of the formation of the horizontally-oriented PV dipoles due to heating in the presence of background shear is the relevant mechanism in this case due to the following two reasons: On the one hand, a recent study by Harvey et al. (2020) has theoretically deduced that the vertical components of the PV tendency equation cannot form negative PV (only decrease PV). Thus, the horizontal components are essential to form absolute negative-PV air (in contrast to just a negative PV anomaly). On the other hand, the analysis of the standard deviation of PV for the convective updrafts shows that the standard deviation is largest in the vertical column directly centered at the convective updraft (Fig. 2) in this document), where the ascent strongly modifies PV and large-amplitude PV anomalies occur. This suggests that there the PV modification is strongest, because small differences in the shape, widths, and intensity of the individual convective updrafts and small spatial displacements of the exact position of the individual PV dipoles result in a large standard deviation (see also reply to specific comment 1). In contrast, if the PV dipoles were to be formed by the cross-frontal advection of vertical PV dipoles, we would expect also a larger standard deviation in the region where the PV poles are supposed to be advected (i.e., more to the right and left of the convective updraft). Moreover, we are not sure, why the cross-frontal advection of vertical PV dipoles is supposed to form the horizontal PV dipoles and does not further distort the shape of the PV anomalies.

**Specific comments**

1. In addition to calculating the composite mean maps, have the authors analyzed the variance? Variance maps could establish whether the mean maps are robust. For example, where the composite mean amplitude is large but the variance is low, the mean fields could be considered robust.

**Reply** We computed the standard deviation (Fig. 2 in this document), however, interpret the signal differently. We agree, that theoretically, one would expect a lower standard deviation for a robust signal. However, in the high-resolution 2-km simulation, the PV field is extremely patchy and fragmented (e.g., Figs. 9e and 10b in the manuscript); each individual PV dipole is slightly different, and only a small displacement of the exact convective WCB trajectory position in the vertical or the horizontal can result in large differences in the grid-point PV field between two individual convective PV dipoles (see also reply to general comment 2). We rather assume that the enhanced standard deviation directly at the center of the convective ascent is an indication that there very strong and localized PV gradients arise due to the localized diabatic heating. We think that generally one can expect a stronger standard deviation of PV in the region where the "action takes place", i.e., where strong diabatic PV modification occurs. In contrast, the surroundings of the convective ascent are characterized by lower standard deviation as only weak PV modification takes place. Thus, we think that the presence of the distinct coherent PV dipole field centered around the convective ascent - despite the very fragmented instantaneous PV fields - is a robust signal. In particular, because at such a high resolution each individual convective updraft (which each also vary in shape and size) is accompanied by a slightly differing PV structure, resulting in increased gridpoint variability.

Figure 2: (a,b) Northwest-southeast oriented vertical cross-section composite of PV (blue and red lines at 0 PVU and 2 PVU, respectively) and its standard deviation (colors, in PVU) for (a) convective WCB trajectories 30 minutes and (b) slantwise WCB trajectories 60 minutes after the start of the fastest ascent.

2. Figure 2 gives the impression that there is a large separation in time and space between the slantwise and convective particles, but the figure only shows the times and locations of maximum ascent. At any fixed time, would these groupings of particles occupy distinct regions of the WCB, or are they distributed more uniformly?

**Reply** The convective WCB trajectory positions at the presented times in Fig. 2 are directly embedded within the regions where also slantwise WCB ascent takes place. Unfortunately, this is difficult to see as convective and slantwise WCB trajectories directly coincide frequently (lines 241 ff. "Despite the differing ascent behaviour between both WCB categories, the convective WCB ascent is directly embedded in the region of large-scale ascent in close proximity to the more slowly ascending WCB trajectories, indicating that although their ascent rates differ, both WCB categories are not distinctly spatially separated"). However, the slantwise WCB trajectories occupy in general a wider region in the warm sector than the convective WCB trajectories and they occur during a longer time period (see Fig. 1b). For clarification, we added the mean latitude of convective and slantwise WCB ascent in the manuscript ["The convective WCB trajectories start their ascent on average slightly further southward at the cold front  $(45.2^{\circ} \pm 3^{\circ})$  compared to the slantwise WCB trajectories whose ascent region extends further poleward  $(47.7^{\circ} \pm 4^{\circ})$ , but the overall region of origin overlaps (Fig. 2) and the convective WCB ascent is indeed embedded in the region of slower WCB ascent."]. After the start of the fastest ascent, the convective WCB trajectories ascend very rapidly through the entire troposphere, i.e., they reach the upper troposphere further south compared to the slantwise WCB trajectories. Thus, with respect to the first arrival in the upper troposphere, the location of convective versus slantwise WCB ascent differs.

3. Does the 2000 to 7000 split in the number of convective versus slantwise particles imply that 3.5 times more mass ascends slantwise?

**Reply** All WCB trajectories should approximately represent equal mass. However, it is difficult to state explicitly that 3.5 times more mass ascends in a slantwise than a convective way, due to the selection criteria we applied. We specifically selected the 10% fastest ascending WCB trajectories for the convective category (cf. section 2c), while for the slantwise WCB trajectories we selected trajectories with ascent rates between the  $25^{th}$  and  $75^{th}$  percentiles. Consequently, a subset of WCB trajectories that ascends rapidly (but more slowly than the 10% fastest ones) exists, and furthermore, the slowest ascending WCB trajectories (which are also considered as slantwise) are also not considered in the composite analysis. However, the compositing technique requires the selection of trajectories with a coherent ascent behaviour to not smear out the signals.

4. In Figure 5c, it is very difficult to distinguish the thick black lines from the thick blue lines.

**Reply** Thanks for the comment, we adjust this figure (see also reply to comment 50 to Florian Pantillon).

- 5. Line 277. Should SWC and RWC be swapped? **Reply** Yes, thank you very much for spotting this typo!
- 6. Line 426. "is" → "are"? (or make "hydrometeors" singular?)
  Reply We corrected this, thank you.

**2 Response to Florian Pantillon**

**Comments to the Author**

**General comments**

1. General comment 1

Unlike the systematic analysis of convective ascents based on composites, the impact on the large scale is investigated based on a single convective PV dipole band, is rather qualitative and uses offline instead of online trajectories. Either extend toward a more quantitative framework, e.g. also based on composites, or at least carefully discuss the results and implications throughout the paper (including in title and abstract) considering these limitations.

**Reply** Thanks for the comment. The conclusions drawn from the illustration of the large-scale impact is derived from the analysis of several of such larger-scale PV dipole bands, which are consistently approaching the waveguide and coincide with an accelerated jet. We did not specifically mention this in the submitted manuscript, but will mention this in the revised version. Moreover, we will also discuss these limitations more carefully. Two additional examples of the propagation of these PV dipole bands are provided here for your information (Figs. 3 and 4 in this document). We think that for the larger-scale PV dipoles a composite analysis is not meaningful, because (i) the PV dipoles all have a different size, (ii) they occur at differing distances to the jet, and (iii) the propagation towards the waveguide strongly distorts the negative PV features which results in different shapes and sizes of the negative PV features. Thus, a composite analysis would smear out the signals, especially the formation of sharp boundaries, such as the enhanced isentropic PV gradient.

2. General comment 2

The "big picture" is diluted in the introduction: the contrast between small and large scale is clear but what is referred to as mesoscale? There is a confusion between isolated cells and organized convection, and embedded convection needs a proper definition (from the example in Fig. 8 it appears to develop along the cold front only). Citations need to be revised and the organization should be improved (see next comment).

**Reply** Thanks for your suggestion, we restructured the introduction, revised the references, and tried to be more explicit about the differences between individual convective cells and organized convection. We added an overview figure to show the occurrence of embedded convection (new Fig. 2), which hopefully helps to better understand WCB-embedded convection and its occurrence in this WCB. Moreover, section 2 provides details about the used criteria to select the convective WCB trajectories.

3. General comment 3

Theoretical considerations on PV production need reorganization: they appear too early in the introduction and are largely repeated in Sections 3.4.3 and 3.4.4. Consider moving the detailed PV discussion to a short theoretical section, which could

later be referred to. Some elements may be moved from the discussion section and grouped with either the introduction or the theoretical considerations.

**Reply** Thanks for the comment on the structure of the theoretical considerations. We restructured the introduction following your suggestions and adjust/shorten accordingly the theoretical considerations in the results sections. We still would like to keep a short theoretical explanation in the results section to highlight the similarities between our results and theory. This will, however, be shortened and more concise.

4. General comment 4

Most of the paper is based on composites of Lagrangian trajectories, which is a very interesting approach but may look abstract to the reader; sometimes a figure in geographical coordinates would be helpful. Placed earlier, Section 4 would well introduce the case study and illustrate the different concepts that are developed in sections 3 and 5, thus remove the need to constantly refer to Oertel et al. (2019) and to Fig. 8(d) early in the paper.

**Reply** Thanks for your suggestions. We agree that it is problematic to often refer to Oertel et al. (2019) in the introduction of the case study. Thus, we added a figure in geographical coordinates to introduce the WCB case study and to illustrate the occurrence of embedded convection and the associated upper-level PV field (Fig. 2 in the new manuscript). This also removes the need to refer to Fig. 8d too early in the manuscript. We believe that the placement of section 4 in the beginning is not meaningful because the reader has not yet been explicitly familiarized with the concept of PV dipole formation (see also reply to comment 81). Thus, the identification and detection of the coherent PV dipole features in the fragmented PV field in Figs. 8d and 9b is rather difficult. Moreover, due to the patchy PV structure, the argument of the coherent dipole signature for convective WCB ascent might not be convincing before the composite analysis. In contrast, after the composite analysis, the reader has already been familiarized with the PV dipole structure and the signal arising from WCB-embedded convection has already been established previously. Moreover, the illustrative example serves as an introduction and transition to section 5 ("PV anomalies on a larger scale and relevance for large-scale dynamics"). For clarification, we added a statement at the beginning of section 4 that this example will further be discussed in the next section.

5. General comment 5

The text contains several repetitions and frequently jumps back and forth between figures. In addition, it often refers to Oertel et al. (2019) and other papers to explain results, which creates a confusion between what is expected from previous work and what is actually found here. Please streamline and clarify to improve the flow.

**Reply** Thanks for your comment, we tried to improve the structure and avoid to jump between figures as much as possible. Moreover, the integration of the additional overview figure (see also reply to general comment 2 and 4) will remove the need to refer to Oertel et al. (2019) too often.

**Specific comments**

1. l. 4-5 not sure this sentence is needed

**Reply** We would like to keep this sentence to emphasize that the impacts of embedded convection have not previously been analysed and to highlight the novelty of this study.

- l. 8-10 the sentence somehow suggests that graupel is part of surface precipitation Reply We clarified this by adding "including the formation of graupel in the upper troposphere".
- 3. l. 11 what does "they" refer to?

Reply We replaced "they" by "the convective WCB trajectories" for clarification.

- 4. l. 17 perhaps insist on negative PV values? (not just anomaly) **Reply** We added PV "values" to this sentence.
- 5. l. 17-19 this is speculative, as only one example is presented

**Reply** We specifically included "can". Moreover, we added later in the manuscript that this process can be seen several times in this case study (see also reply to general comment 1): "The formation of these PV dipole bands on either side of elongated convective ascent regions can be observed at various times ahead of the upper-level trough in this WCB case study (not shown)." Besides, the limitations of our analysis are more carefully discussed in the revised manuscript.

6. l. 21-23 this is also speculative, for the same reason

**Reply** This sentence directly refers to the example that is shown in the manuscript ("An illustrative example of such a convectively generated ..."). We also use the word "can" to emphasize that the described process does not necessarily occur in differing synoptic situations.

7. l. 27 "their"?

**Reply** "their" was supposed to refer to "extratropical cyclones". We slightly changed this sentence to "Moist diabatic processes are known to play an important role for the evolution of extratropical cyclones".

8. l. 29 The two references are not clearly related to moist diabatic processes in extratropical cyclones.

**Reply** This sentence was slightly changed (see also comment above) to focus more generally on moist diabatic processes.

9. l. 35 "potentially affect": better "can affect"?

**Reply** We changed "potentially affect" to "can affect".

10. l. 23-25 this basically repeats what is written above; what are broader implications of the study?

**Reply** We added some potential broader implications: "They thus can be dynamically relevant, influence the jet stream and potentially the downstream flow evolution, which are highly relevant aspects for medium-range weather forecast." Additional broader implications are discussed in more detail in the discussion section. In the abstract we would like to focus on the specific results from this study.

- 11. l. 37 The transition from general WCB dynamics to precise PV theory is abrupt.**Reply** Thanks, we agree. We changed the structure of the introduction and moved the PV theory in a separate subsection later in the introduction (see also reply to general comment 2 and 3).
- 12. l. 42 Please define terms in brackets (or omit).**Reply** We added the definitions for all terms.
- 13. l. 46 Please define f, zeta and theta\_dot.Reply We added the definition of theta\_dot; f and zeta are now defined previously in the revised manuscript.
- 14. l. 47-49 This sounds very similar to l. 34-36.

**Reply** Thanks, this has been modified during the restructuring of the introduction.

15. l. 51 The first two references do not mention mesoscale convective systems.

**Reply** Thanks for spotting this mistake, they do not belong there and were removed.

16. l. 52 What are "convective storms"? (vs. MCS above)

**Reply** We use the word "convective storms" (cf. Chagnon and Gray, 2009) to contrast individual convective cells from the larger mesoscale convective systems. To avoid confusion, we replaced it with "at the scale of individual convective cells".

17. l. 58-59 Is this not what has just been stated in l. 53-57?

**Reply** This sentence has been removed.

- 18. l. 61 The horizontal vorticity vector w\_h must be defined when it is introduced first.Reply Thanks, we included the definition of w\_h.
- 19. l. 60-63 This process is not obvious and requires more details. I appreciate it is supported by a schematic but I do not exactly understand what is what in Fig. 1. Can you make the schematic more reader-friendly, e.g. by illustrating why the vectors are oriented as they are, using more explicit colors and referring to them in the text?

**Reply** Thanks for this helpful comment. We adjusted the schematic, revised the text and more explicitly refer to the colors in the text.

20. l. 71-75 There is a confusion between negative PV, the different types of instabilities and their consequences. Schultz and Schumacher (1999) rather discuss conditional symmetric instability, which is another type of (slantwise) instability.

**Reply** It is true that Schultz and Schumacher (1999) elaborate on CSI, however, they also provide an overview of the other types of (dry) instabilities related to negative PV. Nevertheless, we removed this reference.

- 21. l. 79 "observed" is not the appropriate word for model data. Better "found"? **Reply** We replaced "observed" by "found".
- 22. l. 81-93 This general paragraph would rather better appear before the previous, more specific one.

**Reply** We restructured the introduction such that l. 81-93 appear before the detailed PV modification section (see also general comment 3).

23. l. 87 The cited studies use very different types of "remote-sensing data". More specifically?

**Reply** We think that for the purpose of this study it is not relevant to specify the types of remote-sensing data that were used in the mentioned studies. For your information, Binder (2016) and Crespo and Posselt (2016) used radar observations and retrievals from the polar-orbiting CloudSat satellite, while Flaounas et al. (2016) used microwave measurements from the NOAA-18 and 19 satellites and Flaounas et al. (2018) combined microwave diagnostics from several different satellites.

24. l. 89-93 There is a confusion between what impact and which study relates to WCB or convective systems. Furthermore, the link with forecast errors would better fit with the modification of the large-scale flow in the first paragraph.

**Reply** We modified this paragraph during the restructuring of the introduction.

25. l. 94-114 Rather than pointing what has not been done yet and focusing on very precise questions, this part would better motivate the study if it would emphasize what are open questions (e.g., contribution of embedded convection to WCB dynamics), why they are important (e.g., reconcile small- and large-scale views) and how they are addressed here (e.g., convection-permitting simulations of a case study).

**Reply** The introduction section has been restructured and we included some open questions. The applied methodology (e.g., convection-permitting simulations of a case study) is als included in this paragraph.

26. l. 94 MCSs are not just "individual convective updrafts".

**Reply** Thanks, we are aware that MCS are not just "individual convective updrafts". This sentence was supposed to refer mainly to studies actually related to the analysis of individual convective cells (e.g., Chagnon and Gray, 2009; Weijenborg et al., 2015, 2017). However, we realised that in the context this might be misleading. Hence, we added "Previous studies analysed the PV modification by individual convective updrafts and mesoscale convective systems".

27. l. 116 which case study?

**Reply** We added "analysed" case study. Moreover, we added a statement earlier that states that this study analyses specifically one case study.

28. Past and present tenses are often mixed in this Section, please stick to one or the other.

**Reply** We corrected this, thank you.

29. l. 123 is it the same setup or the same simulation?

**Reply** It is the same setup but a different simulation (because this analysis requires a higher temporal resolution of the 3D fields, i.e., every 15 minutes).

30. l. 125 "grid spacing" rather than "resolution".

Reply We replaced "resolution" by "grid spacing".

31. l. 142-158 It is difficult to get a general picture of the cyclone evolution without reading Oertel et al. (2019). A graphical summary would be helpful, e.g. by adding the cyclone track (and the upper-level trough) on Fig. 2(a).

**Reply** Thanks, we agree. Hence, we added a new figure (new Fig. 2) to better introduce the WCB case study (see also general comments 4 and 5) to show the temporal evolution of the cyclone, the WCB ascent and the according upper-level PV structure in geographical coordinates. We will also include the previously mentioned animations (see general comment 2 in the beginning of the manuscript) to show the evolution of the cyclone and the WCB.

32. l. 142 explicit NAWDEX

**Reply** We changed it to "North Atlantic Waveguide and Downstream Impact EXperiment".

33. l. 150 more than 400 or 600 hPa?

Reply The ascent rates exceed 400 hPa and sometimes even 600 hPa in 2 h.

34. l. 156-157 this last sentence is unnecessary

**Reply** We would like to keep this sentence because we think that it is helpful to state that despite the continuous distribution of ascent rates two distinct categories of WCB trajectories were selected for the analysis.

35. l. 163-166 Why combine two criteria? (400 hPa in 1 h and 600 hPa in 3 h) Is a fast ascent (top 10%) sufficient to be considered "convective"?

**Reply** To get a coherent signal in the composite analysis, the selected trajectories are required to show a similar ascent behaviour and cannot diverge too much during their ascent; otherwise any signal along the ascent would be smeared out. Due to the very diverse ascent behaviour of all (convective) online WCB trajectories, we combined the mentioned two criteria to get a coherent signal (i) in the lower troposphere at the start of fastest ascent and (ii) in the middle- to upper troposphere during the ascent and outflow phase.

We think that one distinct and fixed threshold for "embedded convection" has not yet been defined. Rasp et al. (2016) and Oertel et al. (2019) considered a threshold of 400 hPa ascent in 2.5 hours as convective. The ascent rate criteria used for the composite analysis in this study is much higher. We consider these localized and

strongly enhanced ascent rates as embedded convection (especially compared to the much slower slantwise WCB ascent), however, we also believe that there is not a fixed ascent rate threshold.

36. l. 170-172 The description is confusing: temporal evolution of what? is it really the position relative to the cold front? Where is the upper-level trough?

**Reply** We show the temporal evolution of the location of the start of the fastest WCB ascent in the lower troposphere. Shown are geographical coordinates of the WCB trajectory positions for each timestep. We modified the caption for clarification. The cold frontal surface is also approximately shown, i.e., the initial ascent position of the WCB trajectories in relation to the cold front is illustrated. Moreover, we simplified the figure (see also reply to comment 119). Combined with the new Fig. 2 (evolution of the cyclone and WCB), this figure is now hopefully easier to understand.

37. l. 174 behind rather than "ahead of"?

**Reply** The convective activity occurs ahead of the surface cold front (i.e., east of the cold front) in the warm sector (cf. Fig. 3a in the original manuscript).

- 38. l. 176 again, showing the cyclone position would be helpful for the general picture. Reply The location of the cyclone can now be seen in the new Fig. 2 that was included (see also reply to comment 31).
- 39. l. 183-184 what about the vertical coordinate?

**Reply** The averaging of the fields for the composites was performed on the original model levels, i.e., the vertical coordinate of the composites was not changed. This is (only) possible because the selected WCB trajectories (centered relative to the start of the fastest 400-hPa ascent phase) perform a very similar ascent and do not diverge too much in the vertical during their ascent. Hence also the two ascent criteria are based on the fastest 400-hPa and 600-hPa ascent (see also reply to comment 35).

40. l. 188-189 not only the impact but also the environment of trajectories

**Reply** We added this, thanks.

41. l. 190 is this shown somewhere?

**Reply** This can now be seen in the new Fig. 2, which shows the location of (convective) WCB ascent and the according upper-level PV field.

42. l. 193-194 does it mean that circles in Fig. 2(a) are also at 00 UTC 23 and 24 Sep mainly?

**Reply** The maximum number of circles indeed occurs at 00 UTC 23 and 24 Sep mainly, however, also at times between convective ascent takes place (see Fig. 2b in the original manuscript).

43. l. 201-204 "warmer and moister region": is it really warmer? Fig. 3(a,b) does not explicitly show the contribution of theta to theta\_e and Fig. 4(a,b) suggests that the difference in absolute temperature T is due to a different height. Also, some

indication about how much the two composites overlap is needed, either by showing statistical significance on cross-sections at least by giving the standard deviation around mean values.

**Reply** Thanks for noting this. We replaced temperature by potential temperature to overcome the mentioned issues. In addition, we also included the standard deviation for the initial  $\theta$ ,  $q_v$ , and  $\theta_e$  in the manuscript.

44. l. 205-211 apart from highlighting low-level convergence, it seems that these lines do not add any new information to "the region is warmer and moister"; clarify or streamline.

**Reply** In addition to highlighting low-level convergence and upper-level divergence, this figure shows the WCB ascent ahead of the cold front, the strong and localized  $\theta_e$  gradients and their almost vertical alignment relative to the WCB ascent.

45. l. 214-215, 222-223 what about the difference in height? Is it significant?

**Reply** We included the standard deviation for the distribution of WCB outflow heights in the manuscript and find that the differences are robust  $[10 \text{ km} (\pm 1.0 \text{ km}) \text{ versus } 9 \text{ km} (\pm 1.2 \text{ km})]$ . Besides, we performed a Welch's t-test assuming non-identical variances for the outflow heights of both WCB categories. The test suggested that the difference in means between both outflow heights is highly significant (p $\ll 0.01$ ).

46. l. 218-219 can you be more precise about where to find this information in Oertel et al. 2019?

**Reply** It is found in Fig. 6 and section 5.1.2 in Oertel et al. (2019) (this information was added to the manuscript).

47. l. 221 "the observed rapid convective updrafts" and associated references: it sounds like a conceptual description of potential instability but not necessarily of what happens here

**Reply** Thanks, we realised that the references were not ideally placed and moved them to the previous sentence. We think that this processes actually happens in our case due to the combination of large potential instability and quasi-geostrophic forcing for ascent in the same region.

48. l. 222 mention somewhere the different time scale to emphasize the different ascent rate

**Reply** We mentioned in l. 224 that the ascent takes approximately 18 h ["After an initially swift ascent (due to the centering relative to the fastest 400-hPa ascent), the ascent rate decreases and the trajectories perform a gradual slantwise ascent until they reach their final outflow level at on average 9 km height after approximately 18 h"]. Moreover, the figure caption includes an according statement: "Note the different time axis in (a,c,e) and (b,d,f)." Finally, we slightly restructured the results section to emphasize the different rates earlier in the manuscript.

49. l. 226-228 can you be more precise by giving a value of attained height (average+/-std)

**Reply** In the manuscript we added the standard deviations for the outflow heights for both WCB categories (see also reply to comment 45).

50. l. 230 Please design new panels for the time evolution of surface precipitation in Fig.5; panels (c,d) are already very busy and mixing vertical profiles with a scalar value is extremely confusing. It should also be mentioned somewhere that (a,b) are instantaneous values taken at the respective time of max surface precipitation.

**Reply** Thanks for the suggestions; the figure was improved accordingly.

51. l. 236 "comparatively thick cirrus cloud" compared to what?

**Reply** The word "comparatively" was misleading and was removed.

52. l. 237 turquoise contours?

**Reply** Yes, thank you.

53. l. 237-241 it is unclear how the cirrus cloud related to convection, as its core is located well above the composite trajectory; is it due to a fraction of faster-ascending trajectories?

**Reply** The cirrus cloud above the convectively ascending WCB trajectories is not necessarily formed by the trajectories themselves, but due to mass conservations, the air above the convective ascent region also has to be lifted which subsequently leads to the formation of the upper-level cirrus cloud and the locally elevated cloud top. The formation of (in situ) cirrus clouds above the actual WCB trajectories can also be seen for the slantwise WCB ascent. Previous studies showed that the formation of in situ cirrus clouds is a common feature above the WCB [cf. Spichtinger et al. (2005), ACP and Wernli et al. (2016), GRL]. Nevertheless, some of the trajectories also ascend to higher altitudes, which is however not the main reason for the high ice water content above the convective and slantwise WCB trajectories.

54. l. 243-244 "horizontally more homogeneous": can this really be seen in time-height plots?

**Reply** We concluded that, as the slantwise WCB ascent extends over a long distance, the homogeneity in the time-height plot can be also transferred to horizontal homogeneity. As this statement could be misleading, we removed it.

55. l. 246-247 this is interesting indeed, but may it be due to the compositing process, or are there actual profiles where ice water extends above the tropopause level?

**Reply** Thanks for this comment. We removed this statement because we can indeed not rule out that to some extent this signal arises from the compositing process. We checked individual instantaneous cross-sections, which reveal that only a fraction of convective WCB trajectories effectively contributes to the moistening of the lower-most stratosphere. Figure 5 in this document shows an example where the ice water content above the dynamical 2 PVU tropopause exceeds  $0.05 \text{ g kg}^{-1}$ .

56. l. 251-253 this largely repeats what is written above and is thus unnecessary **Reply** We removed this part.

57. l. 255-256 number of trajectories starting their ascent at that time?

**Reply** Yes, we show the number of WCB trajectories that start their ascent and specified this in the revised manuscript. As the maximum surface precipitation occurs approximately 30 minutes and 1 h after the start of the convective and slantwise WCB trajectories, respectively, the time lag between the precipitation and the number of trajectories is at most 1 h.

58. l. 256 is the precipitation area roughly constant, i.e., are variations in Fig. 2(b) due to variations in intensity or in concentration?

**Reply** For the computation of the domain-averaged precipitation, we only considered precipitating grid points (the number of precipitating grid points does not vary very much), i.e., it is predominantly an intensity effect. The pattern is very similar if either the precipitation sum in the domain or the domain-averaged precipitation including non-precipitating regions are considered.

59. l. 257 "Nevertheless": furthermore?

Reply We replaced "Nevertheless" with "In particular".

60. l. 258-259 what is the citation here needed for? Clarify or omit

**Reply** Oertel et al. (2019) also showed that embedded convection can influence the surface precipitation pattern, but with a different methodology. We specified where to find this information in Oertel et al. (2019).

61. l. 278-282 where is this effect seen in Fig. 5(c,d)?

**Reply** Sorry for the confusion, we now corrected/clarified the references. The former Fig. 5 shows the melting level, the 0°C isotherm, and the transition from the solid (SWC and GWC) to the liquid (RWC) phase. The effect on  $\theta_e$  is shown in former Fig. 4.

62. l. 286 "observed PV distribution": more specifically? Avoid "observed" if from model

**Reply** We removed "observed". The resulting PV distribution is discussed in detail in the following paragraphs, hence, it does not need to be specified in this first paragraph.

63. l. 294-295 "in particular" seems to contradict "despite" above

Reply We removed "In particular".

64. l. 301-302 why that time? (maximum precipitation rate?)

**Reply** We chose this time because it coincides with the maximum hydrometeor content in the mid-troposphere (i.e., also strongest latent heat release). Hence, this time corresponds to the strongest PV modification in the mid-troposphere, and thus, to the clearest and strongest PV dipole signal.

65. l. 303-306 mention the different scales in (a,b) or add box of (a) in (b)?

**Reply** The different spatial scales are already mentioned in the caption, and we explicitly state the dimensions also in the text (l. 302-308 in the original manuscript):

"Note the different spatial dimensions for the convective and slantwise WCB trajectories."

66. l. 308-309 "as a consequence" appears to repeat "due to"

**Reply** We think that the direct effect of convective WCB ascent is the stronger and more localised diabatic heating, while the PV modification is a consequence of this. Hence, we would like to keep this sentence as it is.

67. l. 317 Fig. 5(c) does not explicitly show diabatic heating

**Reply** It is true that Fig. 5c does not show diabatic heating but the hydrometeor contents. However, we state in the text that the diabatic heating maximum is associated with the maximum of graupel and snow formation. We slightly modified the sentence to clarify this: "The maximum amplitude of the PV dipole occurs at about 315-320 K (Fig. 6e) and coincides with the diabatic heating maximum associated with the maximum of the formation of snow and graupel (Fig. 5c)."

68. l. 322-324 this statement appears speculative

**Reply** We believe that to a large extent the patchy PV dipole pattern in the instantaneous PV field corresponds to individual convective PV dipoles, as (i) the PV dipoles mostly coincide with the convective WCB ascent region (e.g., Fig. 8d in the original manuscript), (ii) the composites show a coherent signal, and (iii) the analysis of several individual cross-sections through convective updrafts clearly shows the dipole structure (e.g., Fig. 9 a in the original manuscript).

69. l. 323 specify what to look at in Oertel et al., 2019, Fig. A1

**Reply** We removed this reference, because the patchy PV field is now shown in the overview of the WCB case study (new Fig. 2d-f in the revised manuscript).

70. l. 330-332 I do not clearly see the vertical PV dipole expected in this case according to the previous sentence

**Reply** Fig. 6b shows the positive low-level PV anomaly, while Fig. 6d shows the upper-level low-PV air. However, the slantwise WCB ascent does not lead to PV dipoles with a similar extent as the horizontal PV dipoles formed by convection. Hence, the vertical PV dipole is formed by the regions of enhanced positive low-level PV and decreased upper-level PV.

71. l. 345-346 how is this shown in Fig. 7(a)?

**Reply** Fig. 7a shows the enhanced low-level vorticity, not the vortex stretching. As this might be unclear, we removed the reference to the figure.

72. l. 346-350 is it an interpretation or is it really shown somewhere?

**Reply** This is an interpretation based on the enhanced low-level vertical vorticity (Fig. 7a) and the rapid convective ascent leading to vertical diabatic heating gradients due to enhanced hydrometeor formation in the mid-troposphere. 73. l. 351-358 similar to above, is this shown somewhere for the composite or does it refer to the theoretical schematic only?

**Reply** This refers to both the theoretical considerations and our results. In this section we explain the formation of the convective PV dipoles in our analysis. In the revised manuscript we clarify the references to our results, which have not been stated very clearly in the original manuscript.

74. l. 359-366 This largely confirms what is explained in the introduction

**Reply** We agree that this paragraph confirms the theoretical consideration, which is why we would like to keep it. It highlights that in agreement with theory the processes in the convective and the slantwise WCB trajectories differ. Nevertheless, we shortened this discussion in the revised manuscript.

75. l. 368 again, Schultz and Schumacher (1999) mainly discuss conditional symmetric instability

**Reply** We removed this reference here (see also reply to comment 20).

76. l. 382-384 this partly repeats l. 373-375

**Reply** We shortened these sentences to avoid the repetition.

77. l. 395-398 does it occur here? Is it shown anywhere?

**Reply** Unfortunately, we cannot isolate this process, hence, this sentence remains speculative. We clarified this by stating that "In this way convective activity *could* be maintained".

78. l. 402 "is accelerated by" contradicts "hardly exceeds" above

**Reply** We added that the acceleration in the composite analysis is very small. It now reads "is very slightly accelerated" to point out that the direction of the induced wind anomaly points in the same direction as the low-level jet.

79. l. 406 "PV dipoles" plural or singular?

Reply We changed it to "PV dipole" to be more precise.

80. l. 407-409 for comparison, what is the value of the vertical shear?

**Reply** The vertical wind shear amounts to approximately  $2-3 \,\mathrm{m \, s^{-1} \, km^{-1}}$  in the 4-12 km layer. We added a sentence about the magnitude of the wind shear in the manuscript: "In this case, the vertical wind shear vector of magnitude  $2-3 \,\mathrm{m \, s^{-1} \, km^{-1}}$  between 4-12 km height points in the same direction as the upper-level wind vector, i.e., towards the northeast (Fig. 7c)."

81. An illustrative example of WCB-embedded convection. The purpose of this section is unclear at that point, as it mostly repeats ideas developed in the previous section; such an "illustrative example" would better fit early in the paper to motivate the systematic analysis based on composites.

**Reply** We agree that studies often first show one example before proceeding with a systematic analysis and considered the possibility to show the example earlier in

our manuscript. However, we decided to still first show the results of the composite analysis before the example (see also reply to general comment 4), because (i) the described PV dipole structure is more convincing in the composite analysis compared to the one example shown, (ii) it is difficult to clearly identify the relevant pattern in the instantaneous example when it is yet unclear for the reader what exactly to look for, and (iii) it prepares the reader for the following analysis of the larger-scale impact of the negative PV (Section 5). To better bridge sections 4 and 5 and provide a rational for placing section 4 after the composite analysis, we included a sentence in the first paragraph of section 4 that prepares for the following analyses: "Moreover, based on this example section 5 discusses the potential for the interaction of the convectively generated PV dipoles with the larger-scale flow."

82. l. 420 at 09 UTC 23 Sep 2016

**Reply** The time and date were added in the sentence.

83. l. 422-423 the previous section insists on the presence of graupel to distinguish convective from slatwise ascent: display graupel here only? And does it occur along the cold front?

**Reply** We state that graupel is formed during the convective ascent and is absent for the slantwise WCB ascent. However, we are not sure if the formation of graupel is necessarily required in all cases. The localized and dense cloud with increased hydrometeor content, however, clearly shows the presence of a localised convective updraft. In this example graupel is also abundant and exceeds 2 g kg-1 in the mid-troposphere.

84. l. 423-424 "rapidly ascending WCB trajectories": convective WCB trajectories?

**Reply** We use the term "rapidly ascending trajectories" because they meet the convective ascent criterion used by Rasp et al. (2016) and Oertel et al. (2019) of more than 320 hPa in about 2 h (400 hPa in 2.5 h), but not necessarily the strict criterion used for the composite analysis in this study. The very strict criterion required for the composite analysis results in few WCB trajectories for each time step, which is difficult to visualize (the selected convective WCB trajectories for the composite analysis are located within these outlined regions). However, as also mentioned in reply to comment 35, the composite analysis requires very strict criteria to get a coherent signal in both the lower and upper troposphere.

85. l. 427-429 this last sentence mostly repeats what has just been stated

**Reply** We shortened this sentence, however, we would like to keep the statement about the similarity between the instantaneous example and the composite analysis.

86. l. 432 is PV on the original grid or aggregated in the cross-section?

**Reply** PV shown in Fig. 8b is on the original grid and not aggregated.

87. l. 434 PV below -2 PVU cannot be seen with the colour bar; horizontal PV gradients?

**Reply** Thanks, we adjusted the colorbar and included "horizontal" PV gradients (which was missing before).

88. l. 435-436 is the heating maximum shown somewhere?

**Reply** Unfortunately, we cannot output heating rates. However, we use the hydrometeor formation as proxy for latent heating in the convective updraft (see also comment 67). A sentence was added previously for clarification: "The maximum amplitude of the PV dipole occurs at about 315-320 K (Fig. 6e) and coincides with the diabatic heating maximum associated with the maximum of the formation of snow and graupel (Fig. 5c)."

89. l. 436 "lens" without e

**Reply** Thanks, this is corrected.

90. l. 437-438 please motivate the statement and clarify "mesoscale PV dipole"

**Reply** We include this sentence to highlight the agreement with the composite analysis (section 3.4.4 Partitioning of PV anomalies in vorticity and static stability), and now included a short statement for clarification. We also included the spatial dimension of the PV dipole: "Thus, the mesoscale PV dipole pattern with an extent of approximately 100 km across both poles originates predominantly from the spatial variability of vertical vorticity, in agreement with the composite analysis (Fig. 8b and section 3.4)."

91. l. 441 "rapid WCB ascent": convective WCB trajectories?

**Reply** See also reply to comment 84. We specified "rapid WCB ascent" in the caption for the according figure: "WCB trajectory ascent >320 hPa in 2 h".

92. l. 445-446 "which are generated and further enhanced by convective ascent" sounds speculative

**Reply** We can see that the regions of enhanced convergence are characterized by enhanced low-level PV and coincide with convective ascent. The continuous rapid ascent then additionally enhances the low-level PV. We added "which are generated and *potentially* further enhanced by *continuous* convective ascent".

93. l. 459-450 is the thermal wind vector shown somewhere?

**Reply** We realize that the thermal wind vector (which we replaced by "vertical wind shear vector" to be more precise) has not been shown. The vertical wind shear vector is quasi-parallel to the horizontal wind speed in this case study. We added the direction of the vertical wind shear vector in Fig. 7c,d and mention it in the text.

94. l. 458-464 This belongs to the introduction

**Reply** We agree that this paragraph deals with theoretical considerations that could be placed in the introduction. However, the discussion of this detailed concept might appear out of context in the more general introduction. We consider moving this paragraph to the discussion.

95. l. 470 northwest

Reply We replaced "west" by "northwest", thanks.

96. l. 478 is this supported by section 3 (for this case) or by Shutts 2017 (in general)?

**Reply** The presence and spatial scale of the PV dipoles is shown in section 3. We did not explicitly analyse the interaction between these mesoscale PV anomalies with the large-scale flow. Thus, the second part of the sentence is supported by Shutts (2017).

97. l. 479-480 remove "these"

**Reply** Done, thanks.

98. l. 480-483 this sounds as three times the same statement, clarify or streamline; "effective resolution" has a specific meaning for numerical modeling, better avoid

**Reply** We shortened this paragraph and removed "effective resolution".

99. 1. 485 is this the case for all larger-scale PV anomalies, or for the example of section 4 only?

**Reply** We find several examples of larger-scale PV dipole bands that are aligned with the convective updraft region. However, the exact size varies and depends on the shape and intensity of the convective updrafts. We added a short statement that the exact dimensions are only for the given example.

100. l. 486 is the cold front shown somewhere?

**Reply** The cold front is not shown because the figures are already rather busy. We attached a figure (Fig. 6 in this document) that shows temperature at 850 hPa at 09 UTC 23 Sep 2016 to show the location of the PV dipole ahead of the cold front. Moreover, the composite analysis (Fig. 3a) shows that the convective ascent occurs ahead of the cold front.

101. l. 490 this seems to describe a specific feature rather than "PV dipole bands"

**Reply** We changed "PV dipole bands" to singular to clarify that we analysed one specific PV dipole band.

102. l. 491 southeastward

**Reply** We replaced "east" with "southeastward", thanks.

103. l. 492 repetition of earlier statements

**Reply** Thanks, we removed this sentence.

104. l. 493-497 what is seen where? (which contour, colour, panel)

**Reply** The considered PV feature can be seen in Fig. 11 (blue contour with pink shading). We clarified this in the revised manuscript.

105. l. 502-509 more arguments are needed to support that the convectively-produced PV dipole in Fig. 10(a) evolves into the anticyclonic anomaly in Fig. 10(e): the trajectories spread over a much larger area than this specific feature at 18 UTC, and other PV structures exist during the evolution

**Reply** The analysis of the PV field with hourly resolution clearly shows how the convectively produced negative PV band evolves in the specific feature at 18 UTC. The evolution of hourly fields actually allows for specifically tracing the evolution of all present larger-scale PV features and enables their distinction. However, we think that it is not necessary to show all timesteps in the manuscript, which would require a lot more panels. Moreover, at 18 UTC, the region of negative PV is still largely covered by trajectories (pink dots), indicating that to a large extent the air mass inside this region originates from the negative PV region at 09 UTC. The trajectories indeed spread over a much wider region at 18 UTC. There are two reasons for this. First, as mentioned in the text, only about 60% of all trajectories actually maintain their negative PV for that long. Secondly, the trajectories spread over several isentropic levels, while the PV contours are only shown at 320 K. At higher isentropic levels, the negative PV extends further equatorward and covers another fraction of the trajectories.

106. l. 514 why use offline trajectories, while online trajectories better follow convective ascent?

**Reply** For the analysis of trajectories starting in the upper troposphere, offline (in contrast to online) trajectories were considered because the online trajectories were only started in the lower troposphere to obtain a large number of strongly ascending trajectories. The disadvantage of the online trajectories is that the starting region has to be defined a priori and that due to computational costs (memory allocation) only a limited number of online trajectories can be calculated. To obtain a maximum number of WCB trajectories, the online trajectories were only started in the lower troposphere. Hence, online trajectories arriving in the upper toposphere have all performed a deep ascent from lower levels. However, a large percentage of air parcels that gain negative PV are not directly strongly ascending, but experience PV modification through the "remote effect" of localised heating (note that  $\nabla_h \theta$ is relevant for PV modification, which extends beyond the most strongly heated region; see also section 5.2) as they pass the left side of the convective updraft regions in the upper troposphere. Section 5.2 also shows that the largest fraction of trajectories that gain negative PV are advected quasi-isentropically and pass the left side of the convective ascent region (where the heating maximum is located; see also reply to comment 109). Hence, the number of available online trajectories in the target region in the upper troposphere is too small. Moreover, the online trajectories can only be computed forward, and do not allow for an analysis of their origin. Finally, as the majority of trajectories started within the negative PV region does not ascend directly within the convective updraft, we assume that the offline trajectories approximately represent the actual path of the air parcels. We agree, however, that for strongly ascending trajectories the online trajectories better represent the actual air parcel path than the offline trajectories.

**107. l. 517-518 this largely repeats the previous sentence**

**Reply** We removed this sentence.

108. l. 525-534 the paragraph contradicts the last sentence in l. 523-524 and is confusing altogether; please clarify

**Reply** In the revised version of the manuscript we will rewrite this paragraph and clarify the content.

109. l. 538 how exactly do parcels "gain negative PV"?

**Reply** What we mean is that air parcels gain "negative PV" as they pass the left side of a convective ascent region (which is strongly heated and represents a local horizontal heating maximum), where the diabatic heating gradient resulting from the localised convective ascent is antiparallel to the horizontal vorticity vector, which leads to PV reduction, and eventually negative PV in regions adjacent to the convective ascent region (cf. PV reduction to the left of the updraft region in Fig. 1). We clarify this in the revised manuscript.

110. l. 544 indeed, a comparison with online trajectories is needed to support this result; but again, why use offline trajectories here?

**Reply** Unfortunately, not enough online trajectories are available for a comparison (cf. reply to comment 106). Moreover, we assume that offline trajectories with 15 minute resolution are capable to approximately follow the larger-scale flow. As we use an average over more than 40 000 trajectories, we think that statistically the evolution of the number of trajectories with negative PV is a robust result, which also agrees with the long maintenance of the negative PV in the isentropic PV fields. Also because the majority of these trajectories does not ascend directly within the convective updrafts, we think that offline trajectories with a temporal evolution of 15 minutes are an appropriate approximation.

111. l. 565 what should be compared between these figures? (which contours)

**Reply** We removed this reference, as it is unclear.

112. l. 569-570 this is not sufficiently supported; develop or omit

**Reply** As this conclusion is not essential for this study, we omitted this last sentence.

113. l. 575 not only one case study but one single PV dipole within a cyclone; a first step would be to look at other structures within this cyclone

**Reply** To conclude with this statement in the discussion, we indeed analysed several of these PV dipole bands that occur in this WCB case study (see also reply to general comment 1). Unfortunately, we did not explicitly mention this in the submitted manuscript, but we included such a statement in the revised version ["The formation of these PV dipole bands on either side of elongated convective ascent regions can be observed at various times ahead of the upper-level trough in this WCB case study (not shown)."]. Moreover, we attached two more illustrative examples for your consideration (Figs. 3 and 4 in this document).

114. l. 568 avoid "observations" if model-based

**Reply** We removed the word "observations".

115. l. 586-594 these various impacts of embedded convection appear speculative; please clearly distinguish between what is due to convectively-generated PV anomalies and

to the WCB outflow in general, and be precise about what the cited studies have shown

**Reply** We revised the discussion section and more carefully stated what the cited studies analysed.

116. l. 607 heating is also parameterized, even at convection-permitting resolution, through the microphysical scheme

**Reply** Thanks, we specified this and added "localized heating from the convection parameterization scheme".

117. l. 611-612 "a horizontal resolution of at least 10 km would be required to resolve the convective updrafts": rather a grid spacing of a few km mostly, as in your simulation

**Reply** We changed this sentence and replace resolution by grid spacing: "a horizontal grid spacing of approximately 2 km would be required".

- 118. Figure captions: "shading" better than "colours"Reply We would like to keep "colours".
- 119. Providing titles to subfigures would be helpful, as most display rather complex content Fig. 2(a) is too busy: consider showing less trajectories (every second, fifth, tenth, . . .) and one representative, thicker theta contour per lead time. It took me a while to understand what is depicted and I still do not fully see the position of trajectories relative to the cold front.

**Reply** Thanks, we clarified (former) Fig. 2a. In addition with the new Fig. 2, the figure is hopefully easier to understand (see also reply to comment 36).

120. Fig. 4 are these really "Vertical cross-section composites"? l. 186-187 rather refers to "composites of vertical profiles along the trajectories, i.e. time-height sections along the flow"; (a,b) 300, 320 and 340-K isentropes; (c,d) "(moist-adiabatic) lapse rate" rather than "potential instability"; d\_theta/dz or d\_theta\_e/dz?

**Reply** We indeed show "composites of vertical profiles along the trajectories" and changed this in the according captions. Panels (c,d) show moist stability ( $d\theta_e/dz$ ; this was corrected). We labelled the caption as "moist stability", as we associate one particular gradient with "lapse rate".

121. Fig. 5 "As Fig. 3a,b": not really, better explain again; what do RWP, SWP, RWC, SWC, . . . stand for? Check units; plot (a) box on (b) for comparison?

**Reply** We adjusted the caption and simplified the figures.

122. Fig. 8 this figure does not meet the otherwise high quality standard of the paper: tickmarks are too small and need °N/°E to indicate geographical coordinates (in contrast to km in composites); white contours are hardly seen on panels (b-d); vectors and vector legends are too small on (c-d); the colour bar is not adapted to the noisy field in (d); colour bars are completely saturated for negative values in (d-f); finally, (a) is not standard infrared imagery, what does it show exactly?

**Reply** Thanks, we have increased the quality of the figure, and more specifically added  $^{\circ}N/^{\circ}E$  tickmarks, increased the vector legend, and changed the color of the

contours. We also adjusted the PV colorbar. Panel (a) shows cloud data from the IR10.8 channel; data are obtained from EUMETSAT and plotted to highlight the large-scale cloud band.

---

## Author Response (AR2)

**Manuscript wcd-2019-3**

**'Potential vorticity structure of embedded convection in a warm conveyor belt and its relevance for the large-scale dynamics'**

Oertel, A., Boettcher, M., Joos, H., Sprenger, M., and Wernli, H.

**Response to the reviewer**

We would like to thank Florian Pantillon for his second review and the detailed feedback to the manuscript. We tried to address all of his minor suggestions in the following.

**1  Second review by Florian Pantillon**

**General comment**
The paper has greatly improved in structure and clarity since the first review and my previous concerns have been addressed thoroughly. I appreciate the revised discussion and new schematic, which will certainly also help the reader understand the detailed processes discussed otherwise. Besides a few specific comments and suggestions listed below to further improve clarity, I therefore recommend the paper for publication in Weather and Climate Dynamics.

- l. 25-27 I agree with the conclusion ("our results imply that a distinction between slantwise and convective WCB trajectories is meaningful") but I think the given reasons are too specific and merely repeat what is stated above. Could you give a more general message?

  **Reply** Thanks, we removed the last part of the sentence to avoid the repetition. More general messages can be found in the discussion section and, in our point of view, are not relevant for the abstract.

- l. 47-51 I am still not fully happy with this paragraph: WCBs and convection are put on the same level but they do not generally occur together and the cited studied do not all treat both.

  **Reply** In this paragraph, we actually refer to WCBs and convection individually. Indeed, the mentioned studies mention either WCBs or convection, which both individually can modify the PV distribution and the larger-scale flow. We have tried to clarify this paragraph. Moreover, in the last sentence we had explicitly stated that both systems can **individually** lead to forecast busts.

- l. 165-167 "with a minimum slp of 975 hPa" should appear at the end of the sentence, after "where it becomes stationary"

  **Reply** Placing "with a minimum slp of 975 hPa" at the end of the sentence would be wrong, because the cyclone reaches its minimum SLP during 23 Sep just before

it becomes stationary over Iceland on 24-25 Sep. For clarification we included "The cyclone with a minimum sea level pressure of 975 hPa *on 23 Sep* is located below an upper-level trough and propagates eastward across the North Atlantic toward Iceland, where it becomes stationary on 24-25 Sep". See also reply to comment on Fig. 3.

- l. 278-281 This is not obvious to see (where exactly?)

  **Reply** For clarification, we included the height levels, where the mentioned processes occur. This paragraph now reads: "Following the ascent of the convective WCB trajectories from 1 km to 4 km height (Fig. 5a), below and near the melting level in the vicinity of the 0°C isotherm, i.e., where a transition from the solid (SWC and GWC) to the liquid (RWC) phase occurs (Fig. 4c), $\theta_e$ decreases along the ascent (Fig. 5a) due to melting of snow and graupel falling into the ascending air parcels. At higher altitudes, i.e., following the trajectories above the melting level at 4 km height, $\theta_e$ increases again due to the additional heat release in the ice phase (Fig. 5a)".

- l. 337-340 Repetition of l. 332-334

  **Reply** Thanks for pointing out the repetition, this sentence was removed.

- l. 352 Referring to later Fig. 9d is unnecessary here

  **Reply** The reference to Fig. 9d was removed.

- l. 440 "stabilize" is not the most appropriate word to maintain a convective cloud

  **Reply** This sentence refers to a potential *stabilization* of the convective cloud *against rapid advection with the upper-level flow* due to a deceleration of the wind speed in the center of the PV dipole. We do not explicitly state that this is a mechanism that could maintain convective activity, because we cannot explicitly show that this deceleration also maintains convective ascent. For clarification we changed the sentence to: "The superposition of these two flows leads to a deceleration of the flow in the center of the PV dipoles and potentially reduces the rapid dispersion of the convective cloud with the upper-level flow".

- l. 465 avoid referring to Fig. 10a yet

  **Reply** Thanks, the reference was removed.

- l. 466 Fig. 9b, c?

  **Reply** Thanks, it is also shown in Fig. 9b, hence, we included the reference to Fig. 9b.

- l. 467 red contours?

  **Reply** The red contours show the 2-PVU contour at 320 K (see caption for Fig. 9a), while the rapidly ascending WCB trajectories are outlined in white in Fig. 9b,c. The reference to Fig. 9a was originally included for orientation as it shows a larger extent (position of embedded convection relative to the upper-level trough), however, we realized that the reference to Fig. 9a could be confusing, and hence removed it.

- l. 494-496 The comparison is misleading as scales differ between Figs 1 and 7c (individual convective updrafts) and Fig 9c (aggregated PV anomalies)

  **Reply** In this paragraph we wanted to emphasize the *consistency* between the PV dipole signal at the different scales. We are aware of the scale differences and now also mention this in the manuscript.

- l. 513-514 Repetition of l. 504-507

  **Reply** We substantially shortened the first sentence, but kept a shorter version of it as a transition into the next sub-chapter.

- l. 527-528 Can you quantify the wind acceleration or is it rather a qualitative statement?

  **Reply** We quantified the acceleration with the 2-h circulation anomaly, which for the mentioned time amounts to approximately 5-10 $\mathrm{m\,s^{-1}}$. Additionally, we inverted the absolute vertical vorticity at $320\,\mathrm{K}$ within the negative PV region to obtain the non-divergent wind attributed to vertical vorticity in this region (cf. Oertel, 2020, Chapter 8, PhD thesis), which resulted in a very similar magnitude and direction as the 2-h wind anomaly. The latter is, however, not mentioned or used in the manuscript.

- l. 552-553 Is this shown somewhere or is it an assumption?

  **Reply** Sorry, this is not explicitly shown in the manuscript. Similarly to Fig. 11, we analysed the location of the backward trajectories that are located within the negative PV feature at 09 UTC. The conclusions are based on these figures. We added a *not shown* in the manuscript. For a detailed description of this process see Oertel (2020, Chapter 4.5, PhD thesis), which is now also referenced in the manuscript.

- l. 561-562 Same comment as above; refer to Fig. 11a?

  **Reply** We have previously added a reference to Oertel (2020); see also reply to comment above. The mentioned sentence refers to $t$=-1 h, which is not shown in Fig. 11a ($t$=0 h), although the PV structure looks very similar at both times, which is now clarified in the manuscript.

- l. 564 I do not understand where the values +/-10 pvu come from: these are extremes only and not necessarily related to trajectories that ascent from lower levels.

  **Reply** Thanks, the way we expressed the sentence was not precise. The low-level trajectories have mainly PV values within the range of $\pm 10\,\mathrm{PVU}$. We checked the distribution of PV values of these low-level trajectories. The sentence was adjusted accordingly: "have mainly PV values within the range of $\pm 10\,\mathrm{PVU}$ before they ascend to the upper troposphere".

- l. 586 Fig. 12b?

  **Reply** Yes, thank you very much for spotting this typo! We replaced Fig. 12a by Fig 12b.

- Fig. 2 Consider adding a symbol for the location of Vladiana for clarity

  **Reply** Thanks for this helpful comment, we added a "**L**" to mark the position of cyclone *Vladiana* (see new Fig. 2).

- Fig. 3 The location of the surface cyclone does not appear to match the description at l. 166-167: "propagates eastward and northward across the North Atlantic toward Iceland, where it becomes stationary"

  **Reply** Thank you, we removed "northward", as the total track is indeed only very slightly displaced northward. Moreover, we added "where it becomes stationary on 24-25 Sep" for clarification. This later stage of the overall cyclone evolution is not shown in Fig. 3.

[revised manuscript text omitted]